# Recent Desalination Technologies by Hybridization and Integration with Reverse Osmosis: A Review

**Jhon Jairo Feria-Díaz [1,2], Felipe Correa-Mahecha [3], María Cristina López-Méndez [1,*], Juan Pablo Rodríguez-Miranda [4] and Jesús Barrera-Rojas [5]**

[1] División de Estudios Posgrado e Investigación, Tecnológico Nacional de México/Instituto Tecnológico Superior de Misantla, Misantla 93821, Mexico; jhon.feria@unisucre.edu.co

[2] Facultad de Ingeniería, Universidad de Sucre, Sincelejo 700001, Colombia

[3] Facultad de Ingeniería, Fundación Universidad de América, Bogotá 111321, Colombia; felipe.correa@profesores.uamerica.edu.co

[4] Facultad del Medio Ambiente y Recursos Naturales, Universidad Distrital Francisco José de Caldas, Bogotá 11021, Colombia; jprodriguezm@udistrital.edu.co

[5] Departamento de Energía, Universidad Autónoma Metropolitana/Unidad Azcapotzalco, Ciudad de México 02200, Mexico; jbr@azc.uam.mx

\* Correspondence: mclopezm@itsm.edu.mx

**Abstract:** Reverse osmosis is the leading technology for desalination of brackish water and seawater, important for solving the growing problems of fresh water supply. Thermal technologies such as multi-effect distillation and multi-stage flash distillation still comprise an important portion of the world's desalination capacity. They consume substantial amounts of energy, generally obtained from fossil fuels, due to their low efficiency. Hybridization is a strategy that seeks to reduce the weaknesses and enhance the advantages of each element that makes it up. This paper introduces a review of the most recent publications on hybridizations between reverse osmosis and thermal desalination technologies, as well as their integration with renewable energies as a requirement to decarbonize desalination processes. Different configurations provide improvements in key elements of the system to reduce energy consumption, brine production, and contamination, while improving product quality and production rate. A combination of renewable sources and use of energy and water storage systems allow for improving the reliability of hybrid systems.

**Keywords:** seawater; reverse osmosis; renewable energy; specific energy consumption; integration; hybridization

## 1. Introduction

Although about 70% of the Earth's surface is covered by water, just 2.5% is fresh water [1], and it is estimated that only 1% of this is easily accessible [2]. 40% of the world population currently lives in arid areas or islands where fresh water is scarce [3]. Additionally, an increase of droughts worldwide, resilience reduction to climate change from conventional water resources, and overexploitation have increased dependence on desalination technologies, whose implementation is affected by economic, environmental, technical, social, and political factors [4]. Recent inclusion of water in the stock market is an example of the challenges to be faced in the 21st century. It is estimated that by 2025 two thirds of the world population will face shortages of this resource [5], for which governments must establish functional policies addressing social concerns on water access by the poorest communities [6], while guaranteeing the resource for industrial and household purposes.

The 2030 agenda of the United Nations seeks to: "guarantee availability and sustainable management of water and sanitation for all" [7]. Different water management strategies, along with decarbonized desalination and improvement of irrigation systems, are

key elements to achieving this goal of sustainable development [8]. There is a wide diversity of technologies for desalination and treatment operations for distinct water types. Energy consumption is shown in Table 1 according to the used type of source. Reducing energy consumption is one of the focuses of researchers [9].

**Table 1.** Energy consumption for different water sources.

| Water Supply Alternative | Technology | Energy use (kWh/m³) | Reference |
|---|---|---|---|
| Conventional treatment of surface water | Physical treatments; coagulation | 0.2–0.4 | [10] |
| Water reclamation | -- | 0.5–1.0 | [10] |
| Wastewater treatment | Filtration, coagulation and / or biological treatments | 0.2–0.67 | [11] |
| Indirect potable reuse | -- | 1.5–2.0 | [10] |
| Brackish water desalination | BWRO | 0.8–2.5 | [12] |
| Water Desalination of Pacific Ocean Water | SWRO | 2.5–4.0 | [10] |
| Seawater | SWRO | 2.58–8.5 | [13] |

Source: Reproduced from [10–12].

Seawater and brackish water desalination is one of the most promising processes to solve the world's water shortages. Its use has increased 6.8% per year in the last decade, equivalent to an annual addition in fresh water production of 4.6 million m³/day and, although plants with the highest capacity are scarce, they contribute to most of the installed capacity [14]. According to the International Desalination Association (IDA), desalination plant capacity in the world reached 99.8 million m³/day in 2017, with some 18,500 plants installed in 150 countries [15]. In China alone, some 142 plants operating with seawater were installed during 2018 [16]. Between January 2019 to February 2020, some 155 new plants were put in operation, increasing the installed capacity by up to 5.2 million m³/day [14].

A total of 61% of the world's installed desalination capacity corresponds to seawater, while 30% to brackish water. Moreover, 62.25% is used for municipal purposes and 30.2% for industrial use. The Middle East and North Africa contribute 47.5 % to the installed capacity, while East Asia and the Pacific add up to 18.4%, and North America, 11.9% [17,18]. It is estimated that the world desalination market will grow at a speed of 9% in the coming years. 74% of this growth will come from Europe, the Middle East, and Africa [19].

Currently, more than 20 different technologies are used for seawater desalination [20]. Nonetheless, only a handful of these dominate global water production. Commercial desalination processes are grouped into three broad categories: thermal processes, membrane separations, and emerging technologies. Within thermal processes, multi-effect distillation (MED) and multi-stage flash distillation (MSF) stand out. Reverse osmosis (RO) is the dominant technology [16] in membrane processes, while multi-effect desalination (MED) covers 7% of the installed capacity, and multistage flash systems (MSF) cover 18%, being the most used technology in large-scale dual-purpose desalination plants (water and electricity). On the other hand, RO technologies are more used at the middle and small scale in single process plants (fresh water production) [9], being the ones with the highest participation with 69% of the installed capacity in the world. Other membrane technologies such as nano filtration (NF) contribute 3% to desalination, while electrodialysis (ED) 2%, and reverse electrodialysis (EDR) 1% [18]. In the Middle East, thermal desalination

technologies continue to be overriding, due to their integration with power plants and useful life of more than 30 years [21]. Hybridization of thermal desalination technologies with renewable energy sources is one of the trends in research and development that seeks to improve the sustainability of these processes [22].

Other emerging technologies such as capacitive deionization, freezing, humidification-dehumidification, and desalination with gel hydrates are in the preliminary stages of research and development, and have not reached enough maturity for extensive use. Ultrafiltration, nanofiltration, and ionic filtration are often used as technologies for pre- and post-treatment processes [10,14], being used in small-scale plants and providing less than 2% of the installed capacity in about 940 plants around the world, mainly of electrodialysis and reverse electrodialysis [17]. Electrodialysis is used in industrial applications for selective removal of ions and for brackish water desalination [23]. Desalination through carbon nanopores, inspired by biological aquaporins, is found in a laboratory phase and is a promising technology for desalination [24].

Desalination by means of RO is one of the most efficient and economically viable processes. Recent improvements in flow devices and energy recovery, along with advances in materials for membranes, hybridization with other desalination technologies, as well as its integration with renewable energies, and optimization with artificial intelligence, are some of the most relevant aspects introduced in this document.

## 2. Desalination Technologies with Membranes

Due to the high values of the vaporization enthalpy of water, desalination technologies with phase changes consume substantial energy amounts. Producing 1000 m³/day of fresh water with thermal technologies requires about 10,000 tons of fossil fuels per year [25]. Rises in energy costs and the development of new membrane materials, together with growing regulations regarding greenhouse gas (GHG) emission control have promoted research, development, and establishment of desalination plants that use membrane technology, consisting of a filtration operation driven by pressure differentials, electric potential and/or salinity in which a semi-permeable or selective membrane retains the water components, including ions, and allowing the passage of water molecules [26,27].

Seawater desalination with thermal phase change technologies dominated the market during the first decade of this century. In contrast, the second decade has been characterized by the increase in the use of technologies for desalination of brackish waters, tertiary wastewater, and saline surface waters where individual installed capacities do not exceed 50,000 m³ per day [28]. Currently, about 70% of desalination plants use membrane systems thanks to their high effectiveness and lower energy consumption and costs compared to thermal phase change technologies, being the main supply of drinking water for millions of people [29]. RO is the fastest growing technology in the world, and its market is estimated to reach 9227 million USD by 2022 [19]. Table 2 shows the characteristics of the different membrane desalination technology.

**Table 2.** Comparison of different membrane desalination technologies.

| Technology | Energy Consumption kWh/m³ | Advantages | Disadvantages | References |
|---|---|---|---|---|
| Reverse Osmosis (RO) | 2–6 | Technological maturity. Does not use phase changes. Does not extensively use chemical inputs. Easy to scale and operate. | Membrane fouling and durability. Water recovery reduction with increasing scale. | [30] |

| | | | | |
|---|---|---|---|---|
| | | Does not require elements for energy recovery. Low space requirements. | | |
| Forward Omosis (FO) | 21 | Generation of concentrated brine. Lower environmental impacts. | High energy consumption due to the extraction solution recovery process. | [31] |
| Electrodialysis (ED) | 1–12 | Less susceptibility to scale formation. High salt removal. | High capital costs. Obstruction and loss of energy. | [30,31] |
| Membrane Distillation (MD) | 22–67 | Ability to handle elevated levels of salinity. Low fouling. Use of low-grade heat. | High energy consumption. Low water recovery. | [31,32] |
| Nanofiltration (NF) | 2.54–4.2 | Low operating pressures; high rejection of divalent ions especially sulfates; ability to remove microorganisms and turbidity from water and a fraction of dissolved salts; wide integration capacity as pretreatment for other desalination technologies, achieving cost reduction. | Low ability to reject boron; high levels of membrane fouling | [33] |

According to the salinity degree of the water to be processed, membrane desalination processes can be classified into softening (in which nanofiltration is the core technology), brackish water desalination, and seawater desalination. NF is usually used for water treatment with salinities below 1000 mg/L, with high hardness levels, organic content and bivalent ions. The 'Boca Raton' plant in Florida processes well water with low salinity (466 mg/L) and high hardness values (265 mg/L). To achieve the drinking water quality requirements, it uses a two-stage RO system that produces 151,000 $m^3$/day of fresh water without using acids or anti-fouling agents [34].

Environmental impacts vary depending on the type of used technology and characteristics of feedwater, especially its salinity and temperature. Forward osmosis (FO) has the lowest environmental impacts, followed by electrodialysis and reverse electrodialysis (ED/RED), and operations by membrane distillation (MD) in the third place. While RO showed greater environmental impacts among the studied technologies [35], brine discharges can also impact marine environments, and ground, surface, and underground water sources. GHG emissions are associated with energy use and pollute the air. The energy-water nexus has become increasingly evident, as competition for these resources impacts the reliability indicators, and therefore a balance is required between the two factors to achieve sustainable economic development [36].

### 2.1. Desalination by Membrane Distillation (MD)

Membrane distillation is a second-generation low-grade thermal separation technology, operating at temperatures below 80 °C and with excellent characteristics for seawater desalination operations. It was patented in 1963 and began to gain interest in the 1980s, due to the price reduction of high-efficiency modular membranes [37]. The difference in vapor pressures caused by a temperature gradient across the membrane drives the diffusion of water vapor to the permeate side. Due to the membrane's hydrophobic properties only the vapor can pass through its pores. The permeate flow depends on multiple factors, some associated with the membrane such as pore size and tortuosity, thermal conductivity, and thickness, while others are associated with operating conditions such as the temperature and the flow rate of the feedwater. The specific energy consumptions of MD systems vary from 1 kWh/m³ to 499 KWh/m³, while water costs are between 0.3 US $/m³ and 130 US $/m³, depending on the type of water supply and energy source. Research on this technology is focused on improving energy efficiency, integration with renewable energies, and hybrid systems that make it possible to recover waste heat from different processes in a profitable manner [26]. Figure 1 shows a basic scheme of operation of a Membrane Distillation System.

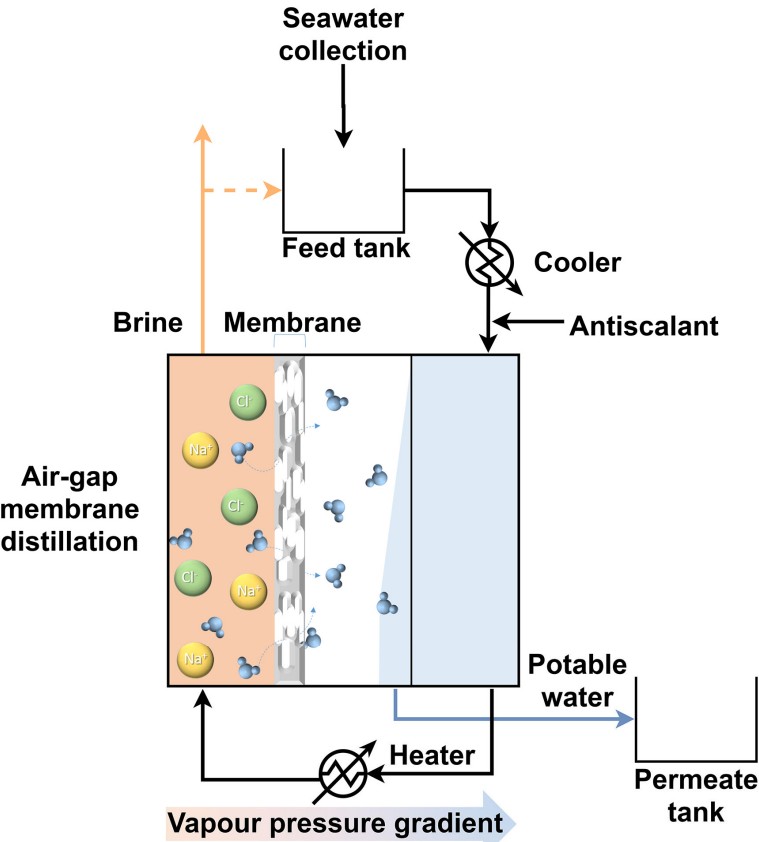

**Figure 1.** Membrane distillation operation with an air-gap configuration taken from [24].

Membrane distillation requires hydrophobic coatings to guarantee selectivity for the exclusive passage of water vapor. The most used compounds are polytetrafluoroethylene (PTFE), polypropylene (PP) [38], polyvinylidene difluoride (PVDF) and erfluorosilanes [39]. Use of nanoporous carbon [40], graphene [41], polystyrene (PS) [42], and Bucky paper has also been researched, in addition to bio-inspired silicon structures, to improve this technology operation for both seawater desalination and other industrial applications

[43,44]. The electrospinning method is a novel process for the synthesis of nanofiber membranes with a high specific surface area, high porosity, high tortuosity and interconnectivity of pores, which achieve rejections of up to 100% of salt [45]. Use of graphene oxide has gained increasing interest as a membrane synthesis material, and its versatility to be functionalized allows obtaining superhydrophobic materials [46].

Membrane fouling is a widespread problem that reduces both useful life and productivity of the system. The use of low power ultrasound (8–23 W) has been researched as a membrane cleaning technique using brine from an RO operation with air-gap membranes, achieving a 100% increase in permeate flow. Thanks to the increase in mass transfer, salt rejections reached values higher than 99% [47].

### 2.2. Desalination by Forward Osmosis (FO)

Forward Osmosis (FO) is one of the most promising emerging technologies for desalination and wastewater treatment operations. Publications on its use have had a growing trend since 2009 thanks to the advancement in new membrane materials. Recirculation of the extraction solution and energy consumption are two of the most-researched restraining aspects [48]. It is also used for leachate treatment with water recovery in sanitary landfills; industrial, agricultural, and urban waters [9,49,50]; recovery of acid discharges [51]; recovery of nutrients from diverse types of effluents [50,51], and liquid mining for the recovery of conventional and radioactive metals and minerals [52,53]. Figure 2 shows a diagram of operation of a Forward Osmosis System.

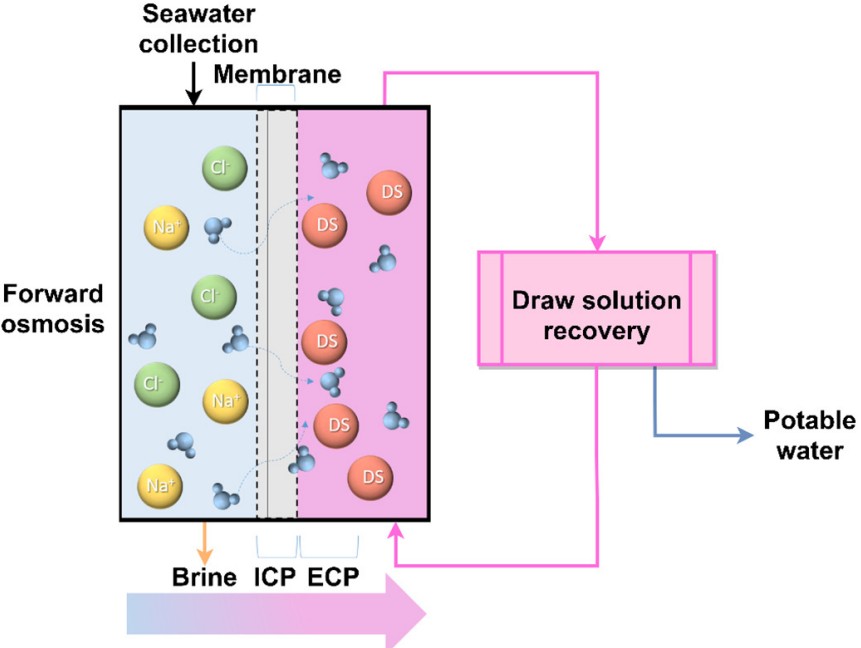

**Figure 2.** Forward osmosis diagram of the areas in which the internal and external concentration. polarizations (ICP & ECP) occur. Taken from [27].

FO uses the differences in osmotic pressure across a semi-permeable membrane, between a concentrated extraction solution (DS) and the diluted feeding solution (FS) [54]. It has been used for food processing, treatment of discharges, desalination, and power generation. Compared to RO, it uses moderate pressures, leading to a lower propensity for scale formation with very high permeate recoveries [48], which can reach 90% [16]. FO has low specific energy consumption, between 0.25 and 1 kWh/m³; nonetheless, the extraction solution requires post-treatment. The most used is thermal distillation that con-

sumes significant energy amounts. Waste heat is generally used for this operation. Combination of technologies, including solar energy, can improve energy yields, and the environmental and economic indicators of the operation [26].

Companies such as OASys, Modern Water, Hydration Technology Innovation, and Trevi Systems have improved their desalination technologies with FO membranes; nonetheless, applications are mainly limited to wastewater treatment in the petrochemical industry. Trevi Systems achieved a technology development for the recovery of the extraction solution using solar energy [10].

In Oman, the Water Plc Company (London, UK) installed the world's first FO desalination plant in 2012. It has a capacity of 200 m$^3$/d. In July 2018, the largest demonstration plant in operation, with a production of 500 m$^3$/d [16], was installed in the Zhoushan Islands of China. Although this technology is very promising, there are still barriers to overcome around costs of the membranes and use of the extraction solution requiring post-recovery treatments, thus making profitability economically unfeasible on a large scale. It is expected that the establishment of hybrid systems powered by renewable energies can overcome some of these barriers [55]. Specific consumption of FO is around 1.95 kWh/m$^3$, and when using hybrid RO-FO systems this can be reduced to 1.83 kWh/m$^3$, while an FO-RO coupling manages to reduce it down to 1.47 kWh/m$^3$ [56].

### 2.3. Desalination by Electrodialysis (ED)

Electrodialysis is an electrochemical process with more than 50 years of industrial development. Its advance has been linked to progress in ion exchange membranes. It does not require use of chemical products in an intensive way and can be coupled to energy sources easily renewable. It can be used in small-scale desalination operations and for environments requiring flexibility such as devices on board vessels and submarines, with 50% recovery, and it is used for ion removal in aqueous solutions and desalination of brackish water. However, due to advances in RO, ED plants for seawater and brackish water desalination have had less and less participation during this decade [16].

Electrodialysis desalination uses a system of different chambers separated by ion-selective membranes which travel from one chamber to another, driven by the potential generated by a direct electrical current applied by electrodes immersed in saline solution. The negative ions pass through the anion-permeable membrane moving towards the anode, while positive ions do the same through cation-permeable membranes moving towards the cathode. Thus, ions are concentrated in some of the chambers, leaving desalinated water in others [57]. This currently receives a lot of attention from researchers thanks to the fact that use of multivalent membranes can eliminate magnesium and calcium. The specific energy consumption for seawater desalination with this technology has been reduced from 6.6 to 3.6 kWh/m$^3$. In recent years, some optimizations of multistage configurations have gone as low as of 2.2 kWh/m$^3$ for artificial waters with NaCl concentrations of 510 nM, a value close to the 1.8 kWh/m$^3$ consumed by state-of-the-art RO technologies with the same type of water [58]. Studies with seawater in multistage configurations achieved values of 3 kWh/m$^3$ [23].

Now, electricity consumption of electrodialysis desalination is mediated by the solution salinity, water electrical resistance, and the system membranes. The temperature increases reduce the electrical resistance at a rate of 2% for each °C, with which reductions in energy consumption are achieved. The same happens in reverse electrodialysis systems (EDR), and for this reason the coupling with photovoltaic thermal collectors could improve this technology operation, by using the heat to preheat the feedwater to the desalination unit [26].

This technology has some challenges to overcome, such as the membrane fouling caused by deposition of anions and organic compounds as well as the selectivity to permeation that have caused difficulties in the scaling process. This is used for the municipal wastewater treatment in the pharmaceutical, chemical, and food process industries. It has a higher percentage of water recovery than RO technologies, is easy to operate with long-

life membranes, allows operations at elevated temperatures, and usually does not require pretreatments or extensive post-treatment [29].

### 2.4. Reverse Electrodialysis (RED)

Reverse electrodialysis was introduced for the first time in 1954 but low performance of the membranes slowed its development. Advances in materials in recent decades have rekindled interest in this technology [59]. It uses salinity gradients for electricity generation. The use of anthropogenic brines increases its electricity production capacity; therefore it has an important potential for industrial and domestic wastewater processes, for desalination processes [60], energy recovery in RO, and conventional electrodialysis systems. Thanks to the use of concentrated brine that allows higher generation rates [61], recent developments using capacitive membrane deionization and direct contact membrane distillation have achieved a 39.0% reduction in energy consumption compared to a two-step RO system [62]. Other applications of this technology include wastewater treatment with simultaneous generation of electrical energy, as well as energy storage [63], which has allowed the development of flow batteries. It is estimated that global energy potential per salinity gradient of natural waters is higher than 27,000 TWh/year. Around 2000 TWh/year can be extracted, 24 h/day. Integration with hydrogen generation could produce 38 Mt/year of this fuel, while the production potential of wastewater discharged to water sources is 18 GW [60].

The first RED demonstration plant was installed in Italy in 2014. It had 125 pairs of cells and 50 m$^2$ of membranes. Using brackish and saturated water, the system achieved a production of 1.3 W/m$^2$, as well as wastewater treatment with the removal of Acid Orange 7 Azo Dye [64]. Subsequent extensions were made to 400 m$^2$ of membranes increasing the power of the system [65]. The largest plant processes up to 10,000 m$^3$ per day and is installed in South Africa [66].

However, there are multiple difficulties that have slowed down large-scale RED industrial applications. It has been estimated that the minimum power to achieve economic viability is 2.2 W/m$^2$, which has not been achieved with natural waters. On the other hand, prices of membranes are well above the commercial range of other technologies such as RO and show fouling problems [28,59].

### 2.5. Reverse Osmosis (RO)

The first RO desalination process was marketed by Loeb & Sourirajan in 1964 [21]. Since then, it has had important advances that have positioned it as the leading technology in desalination operations. It is versatile, thanks to the fact that water evaporation is not necessary for its separation. It has a relatively low energy consumption compared to thermal technologies [67,68], has high flexibility to work under different salinity conditions, takes up little space, and is easy to operate and automate [69,70]. Installed capacity in RO plants is currently more than 60 Mt/day, with an annual growth between 10% and 15%, and a combined energy consumption of 100 TWh/year [71]. RO units are commercially available in varied sizes, from household applications with capacities of 0.1 m$^3$/day, to sizes for industrial and municipal use with capacities of up to 900,000 m$^3$/day [26]. Figure 3 shows a representation of the rolled membranes in common use for seawater desalination processes.

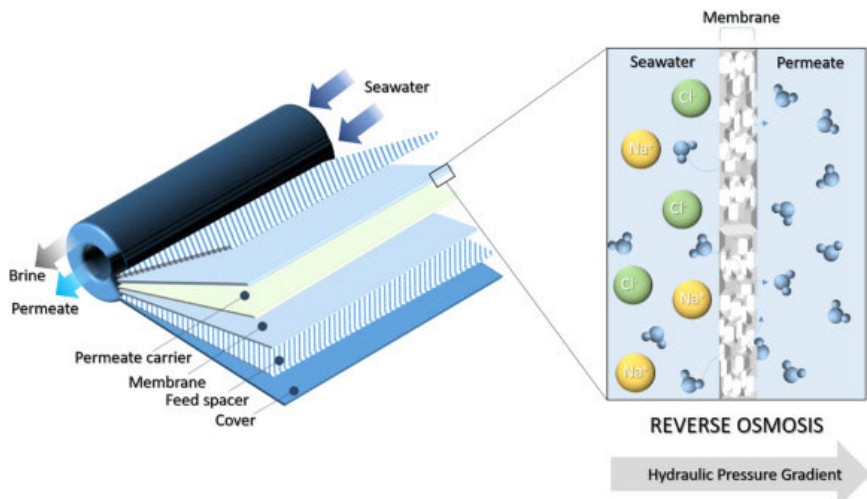

**Figure 3.** Spiral wound membrane and driving force principles (inset) of RO. Taken from [9].

This is currently the most reliable technology for seawater desalination at the lowest capital costs [68] and lower energy consumption; however, it has significant maintenance costs associated with pretreatment operations and membrane replacement that, combined, contribute 25% of the total operational cost [3]. According to the water quality to be processed, RO operations can be classified into brackish water plants (BWRO), with salinities between 500 and 10,000 mg/L and seawater plants (SWRO), with salt contents close to 30,000 mg/L. The efficiency of these plants depends on multiple factors such as operating parameters, membrane type, configuration, and feedwater characteristics [69].

Osmotic pressure is a trait linked to the colligative properties of mixtures and has important impacts on RO systems. High salinities cause increases in osmotic pressure requiring higher operating pressures. Seawater (35,000 mg/L of salts dissolved) have an osmotic pressure of 2413 kPa. For the system to operate correctly, 10,342 kPa will be needed to overcome both the osmotic pressure and pressure drop across the membrane [69]. Therefore, rises in salinity produce higher costs associated with the pumps' operation and increase the concentration of polarization and fouling of the membranes, reducing their useful life [65,67].

The first asymmetric cellulose acetate (CA) membrane for RO systems was developed by Loeb and Sourirajan [72] in the 1960s. Then, in the 1970s, Cadotte et al. [73] developed a new membrane, composed of a thin film composite (TFC), which monopolized the market for many years. Nowadays, and with nanotechnology development, a new type of composite membrane called the thin film nanocomposite membrane (TFN) has been consolidated [74]. Normally, there are two techniques that can be used to synthesize TFN membranes. The first technique, also the most popular, is based on the interfacial polymerization (IP) technique in which nanofillers are embedded within the polyamide (PA) layer during the IP process. The second technique is based on the coating or deposition of the surface in which nanofillers are introduced on the existing PA layer. Lately, a combination of these two techniques has been used for the manufacture of nanofillers of TFN membranes [75].

The obtained water quality by RO is relatively lower (400 TDS) compared to thermal technologies (25 TDS) [3], however, it is considered acceptable for most agricultural, industrial, and human consumption applications. RO can remove all colloidal material and dissolved solids with a size greater than 1.0 nm from a liquid solution. The process normally consists of three stages: pretreatment, RO operation, and post-treatment [27].

The specific energy consumption is also associated with the temperature and water source quality, operating conditions, process scale and energy source, varying between 1.5 kWh/m³ to 15 kWh/m³, while the cost of fresh water can vary between 0.26 US $/m³ up

to 1.72 US \$/m³ [26,69]. According to thermodynamic theory, the minimum consumption for desalination of seawater with 33,500 mg/L of salt and 25 °C (Pacific Ocean conditions), and assuming 50% recoveries, is 1.1 kWh/m³. The company Affordable Desalination Collaboration (ADC) achieved consumption of 1.58 kWh/m³, with recoveries of 42% and flows of 10.2 L/m²h [10]. However, when pre-treatment, distribution, and post-treatment consumptions are included, the consumptions of large-scale plants can increase to values between 3.5 and 4.5 kWh /m³ [76].

Voutchkov and his team [10] estimated the costs of water production and energy consumption of 20 SWRO plants built in the United States. Table 3 shows results for medium and large-scale plants, (a medium plant is considered to have a capacity of 40,000 m³/day of fresh water production). Energy consumption contributes between 25 to 40% of the fresh water cost. The RO system consumes 71% of the energy of the entire plant, so these values must be included in the pre-treatment energy consumption (10.8% of consumption), feedwater pumping (5.3%), fresh water distribution (5%), and the consumption of other storage facilities, maintenance, and disposal of brine (7.6%).

**Table 3.** Typical costs and energy use of desalination systems. Taken from [10].

| Classification | Cost of Water (US \$/m³) | Energy Use (kWh/m³) |
| --- | --- | --- |
| Low-end bracket | 0.5–0.8 | 2.5–2.8 |
| Medium range | 0.9–1.5 | 2.9–3.2 |
| High-end bracket | 1.6–3.0 | 3.3–4.0 |
| Average | 1.1 | 3.1 |

The SWRO specific energy consumption was reduced from 20 kWh/m³ in the 1970s to 2.5 kWh/m³ by 2010 [77], all thanks to improvements in the efficiency of the high-pressure pumps, inclusion and advance in energy recovery of devices, continuous improvement of high-performance membranes, and in the design of the membrane module to reduce the pressure drop through it. However, there is still room for maneuvering to implement improvements [78], hybridization with other desalination technologies, multipurpose plants, and coupling with renewable energy are fields of growing interest that can take this technology along increasing paths of sustainability and resilience.

Seawater desalination costs in the large capacity plants currently installed in the world vary between 0.35 and 1.87 US \$/m³, and in the case of brackish water it is between 0.35 and 1.53 US \$/m³ [14]. In an average SWRO plant, 44% of the costs are associated with energy use [6]. Efficiency of the older plants can only be 10% while the more modern ones reach values of 50%. Motors, pumps and separation systems are the units that contribute the most to system inefficiency [79]. Furthermore, RO membranes have made important advances in the last two decades, achieving water production from seawater and other water at reasonable costs [80]. Since the operating pressures for SWRO are between 49.34 and 67.11 atm [81], the high-pressure pump is responsible for up to 68% of the desalination energy consumption by RO [82]. Advanced exergy analyzes performed on a BWRO desalination plant in the Canary Islands, Spain, found exergy destructions (usable energy), 92.94% in the feed pump, 70.61% in the high-pressure pump, and 7.83% in the RO system. About 198.78 kW of exergy is inevitably lost [83].

Use of Energy Recovery Devices (ERD) has allowed a significant reduction in energy consumption of the RO thanks to the transfer of hydraulic energy from the brine to the feed, reducing consumption of high-pressure pumps. Francis turbines were the first to be used followed by Pelton turbines. Efficiency of these devices is reduced by the conversion of hydraulic energy from the brine to mechanical energy in the device and the new conversion to hydraulic energy in the water [76]. In the 1980s, the development of positive displacement devices, known as isobaric chambers, achieved a considerable increase in the energy transfer efficiency. The DWEER TM from Calder, SalTec DT from KSB com-

pany, OSMOREC, and RO-KINETIC are some examples of ERDs based on positive displacement, while the Pressure Exchanger (PX) from Energy Recovery Inc., and the iSave ERD from Danfoss are examples of commercial isobaric chambers based on rotary displacement technologies used in large-scale plants [84]. PX devices have been reported to have energy efficiencies greater than 95% [76]. There is little research on the development of ERD devices for small-scale plants (production less than 50 m³/day). A study carried out with a new HPP-ERD device showed that with an additional investment of 6.3% in this type of device, it is possible to reduce the fresh water cost produced between 17.8 and 21.9%, as well as reduce the capital recovery period by 12.3%, using ERD devices designed for these low fresh water productions [85].

### 2.5.1. Operation Configurations of RO

Although RO systems are more efficient in desalination of brackish water than seawater due to their lower content of total dissolved solids, the operational configuration of RO plays a very relevant role in the overall efficiency of the desalination system.

There are four widely studied and used configurations in the desalination industry: single-stage RO, series two-stage RO, multi-pass configuration and closed-circuit RO. In single-stage operations (Figure 4a) there is the disadvantage that the increase in pressure required to achieve reasonable permeate flows increases thermodynamic irreversibility and, therefore, the process' Specific Energy Consumption (SEC) is increased [86]. In the configurations of two or more stages, the brine from the first unit enters the second as feed (Figure 4b). This operation makes it possible to maintain a more homogeneous driving force throughout the system, for which the pressure of each stage is conditioned to the osmotic pressure. The osmotic pressure increases in the feeding of each RO stage to reduce energy dissipation, and therefore, SEC. However, the complexity of the system and its capital costs increase [87]. Two-stage RO configurations (Figure 4b) became popular at the beginning of the century for desalination of seawater.

The concentrate from the first stage was fed to a second stage, improving recovery (from 45% to 60%), increasing the capacity of the plant up to 1.5 times, while reducing the amount of concentrated brine by 2/3 and the size of the plant by 2/3. However, this configuration is penalized by higher energy consumption and requires elements that withstand high pressures (10 MPa) in the second stage due to the high salinity of the brine from the first stage. This system has worked in plants in Trinidad and Tobago and in the Canary Islands [88], however this configuration is more often applied in BWRO systems due to its lower pressure requirements and energy consumption [76].

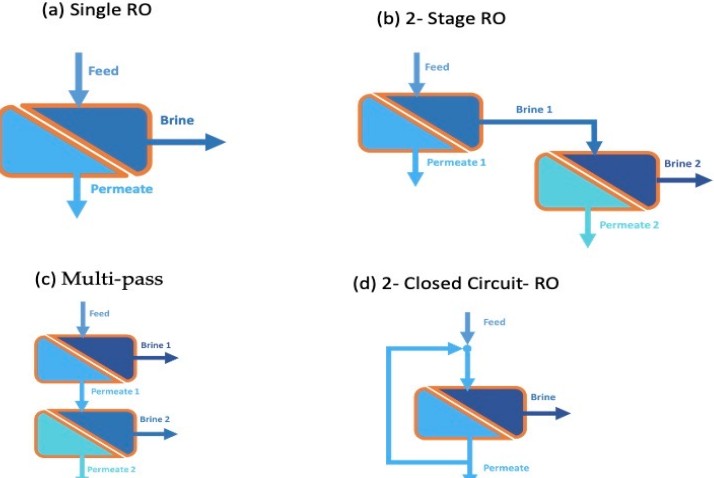

**Figure 4.** Different RO Configurations commonly used in desalination process. (**a**) Single RO, (**b**) 2- Stage RO, (**c**) Multi-pass and (**d**) 2- Closed Circuit—RO.

Recently Ruiz et al. [89] evaluated two two-stage RO configurations to treat brackish water from an underground well, along with their optimal operating points in terms of minimum specific energy consumption (SEC) and recovery of maximum flow rate (R). The authors found that the operating margins were quite wide and were influenced by the hydrochemical fluctuations of the groundwater that fed the system. Therefore, they recommended more flexible designs and configurations of BWRO systems, capable to adapt to the most common and average operating conditions of the brackish water used. In that sense, the authors developed a simple design methodology [90] and a computational algorithm that allows a flexible design and more realistic operating conditions for RO systems [91]. This allows the user to see all practical solutions and choose the most suitable one. Nonetheless, complex optimization methods such as particle swarm methodology, genetic algorithm, and species-conserving genetic algorithm are rarely applied to RO processes of seawater and brackish water. In this sense, greater efforts must be made, especially for multi-objective optimization, to assign the best cost-effective design of a multi-stage RO process under optimal energy and environmental impact options [92].

Tighter regulations regarding water quality have forced scientists to implement continuous improvements in desalination with RO [67]. Multi-pass configurations, in which the permeate improves its quality when fed to subsequent units of RO (Figure 4c) have had special acceptance with the increase in demands of water quality [93]. The subsequent RO units behave like BWRO systems, with lower pressure requirements, although they sacrifice, in part, the specific energy efficiency due to the recovery reduction [94]. Although these systems generate higher-quality water, they are more expensive and produce less fresh water compared to a single pass RO system with the same feed rate.

The operating pressure of the first pass could be reduced by 65% to 75% when comparing the total pressure drop. For this reason, a boosted RO feed pump is required, which inevitably increases the system costs [10].

One-step configurations with partial recirculation of the permeate allow reduction of the operating pressure (Figure 4d) by reducing the feedwater salinity and improving the fresh water quality, achieving lower energy consumption compared to a two-pass system. However, water production is also reduced compared to systems without recirculation [93].

There are other configurations such as Internal Stage Designs (ISD) in which a membrane with high rejection (and high permeate quality) is installed, followed by high flux membranes (and low permeate quality) in the bottom, achieving a lower hydraulic power pressure. With consumption reduction and a longer useful life of membranes [93], there is little research that validates the ISD. An optimization study of the design of this type of element for large-scale processes evaluated 35 commercial SWRO membranes, showing that it is possible to improve recovery rates (increasing up to 8%) and energy efficiency compared to traditional RO. Optimal configuration depends on the fouling potential and the salinity of the feed water [95].

Another strategy to reduce energy consumption of SWRO seawater desalination consists of making dilutions with recycled wastewater, which allows reducing salinity and therefore consumption associated with the systems hydraulic pressure, while recovering industrial, agricultural or urban water. Low operating pressures have positive impacts on fouling reduction, polarization concentration, and useful life of the membrane [48]. Apparently, the search for RO configurations that improve permeate quality with low energy consumption is already well explored [78]. Thus, improvements in RO efficiency will come from novel approaches and its hybridization with emerging technologies of desalination and integration with renewable energies.

When analyzing the application trends of distinctive designs in operation, it can be observed that size of the SWRO plants has been increasing while specific consumption has been decreasing. The main objectives of RO system designs are focused on water quality and reduction of energy consumption. Unfortunately, a lack of coherence in energy studies between different authors has made it difficult to analyze the influence of factors

on energy efficiency [76], so it is recommended to implement standard measurements that allow reporting the values of the key variables of process, the configurations used and the consumption values of both the RO units and the other components in the plant. Exergy analyzes show that improving pump efficiency and minimizing pressure drop across membranes are the main strategies to improve thermodynamic performance [96]. Similarly, mathematical models have been used to research the influences of operational and optimal design parameters on RO performance [97]. Sassi & Mujtaba [98] compared two energy recovery alternatives: using a turbine and using a pressure exchanger. The authors proved that energy recovery through the pressure exchanger has better results, with a 50% reduction in pumping costs.

### 2.5.2. Limitations and Problems of RO

Although RO is a technology with relatively low energy consumption, increases in desalination demand imply use of increasing energy amounts. Due to the current and future polluting nature of energy sources, it is estimated that by 2050 the global emissions related to desalination processes reach 400 million tons of $CO_2$ per year [13]. Countries such as Kuwait consume half of their oil production for seawater desalination and energy cogeneration, with environmental impacts on air quality, abiotic depletion, acidification and eutrophication of waters, depletion of the ozone layer, photochemical oxidation and release of compounds toxic to humans and marine life [99]. Environmental aspects are in the sights of many researchers as they offer important limitations for the expansion of RO.

On the other hand, water treatment with high salinity and brines generates increases the hydraulic pressure required to overcome the osmotic pressure difference between the feed and permeate, which is why the typical recovery in seawater is 50% and up to 60% for RO single stage, a relatively low value compared to thermal technologies. Since MD has a low sensitivity to salinity, it can be used to increase production of fresh water in hybrid schemes with RO [100]. Nonetheless, the chemical substances used in the pretreatments can have a negative influence by generating scale in the MD membranes [101].

Scale formation from less-soluble salts is one of the main restraining factors in terms of flow recovery and is responsible for additional costs in RO systems. Ruiz y Feo [102], for groundwater feeding an RO system in Gran Canaria and Tenerife islands in Spain, found that in some cases the maximum flow recovery was only around 60%, even using a specific silica anti-fouling, having a considerable impact on the process viability.

Another major problem with seawater reverse osmosis desalination (SWRO) is boron rejection. Boron effects can cause health problems for humans and plants [103]. Boron's low retention by commercial RO membranes poses difficulties to achieve the levels required by the WHO and the European Union of 2.4 mg/L and 1.0 mg/L, respectively, since this element is found in relatively high concentrations in seawater. This restriction has become one of the most challenging problems that membrane manufacturers have had to deal with [104].

Studies carried out in Spain with a pilot plant with 7000 L/h feed showed that an increase in temperature reduces the efficiency of boron rejection by the membrane [105], explained by the increase in membrane permeability and diffusion of ions [106].

However, Li et al. [107], using polyamide reverse osmosis membranes under simulated seawater conditions, allowed boron rejection in these membranes to increase up to 93.10%, with a salt rejection rate of up to 99.57%.

The use of multiple steps with increasing pH is the most widely used methodology to reduce boron in the permeate. Hybridization with adsorption technologies may be a feasible solution [108], as it is the use of ion exchange resins and electrodialysis, electrode ionization [109], and direct osmosis [110] processes. Electrocoagulation pretreatment has several advantages in areas where the electricity cost is low [111].

On the other hand, polymeric membranes tend to lose their long-term permeability due to ease of fouling, especially when used in batch systems or powered by solar or wind energy due to intermittent operation, in addition to not being resistant to chlorine [70].

Furthermore, it has been shown that membrane aging, through incrustation and cleaning cycles, has a higher level of degradation when compared to passively aged membranes, exposed only to hypochlorite [112].

Precipitation of inorganic salts, accumulation of particulate material, and biofouling formation are the main causes of RO membrane fouling, increasing maintenance costs, making disinfection operations more difficult, and reducing their life time. Therefore, transport models in RO desalination systems used to estimate flow and decrease in the water permeability coefficient of membranes depend on the system operating time and empirical parameters [113].

Many researchers claim that intermittent operation in RO systems leads to increased rates of membrane fouling [114]. Nonetheless, Freire & Bilton [115] demonstrated that intermittent operation alone does not have a significant negative impact on membrane performance in the short term (several days of operation in a cross-flow unit). According to the authors, use of antiscalant and membrane rinsing with permeate water prior to prolonged shutdown periods may decrease the reduction in membrane permeability. Comparable results were reported by Ruiz & Nuez [116] for a large-scale BWRO desalination plant operating approximately 9 h a day, through a performance analysis performed over a 14-year period. Authors proved that daily shutdowns and start-ups did not cause additional problems at the desalination plant, indicating that intermittent operation of BWRO desalination plants is feasible in the long term.

It is estimated that by 2025 two million residual RO modules will be generated per year [117,118]. Hydrodynamic manipulation and chemical cleaning of the membranes are the most used techniques to remove fouling [119], while feedwater pretreatment is the most widely used technique to prevent it focusing on reducing total dissolved solids, turbidity, silt density index, colloidal particles, and microorganisms [120].

The study of technologies based on electrochemistry has increased as an alternative in scenarios in which electrical energy is affordable. The Electrocoagulation Process (EC) has been researched to be used as a water pretreatment to reduce organic matter in brackish and marine waters. Efficiency in Dissolved Organic Carbon elimination (57.5%) increased with reduction of pH and spacing between electrodes, with optimal values at pH 4, and stirring speeds of 90 revolutions per minute, for 30 min, with a SEC of 0.17 kWh/m$^3$ [121]. Tests carried out for the treatment of oily wastewater improved water quality with reductions of 95.7% of the chemical oxygen demand. Equivalent results were observed for brackish water pretreatment [111]. Hybrid electrocoagulation, electroflotation, and electrodisinfection processes have also been evaluated for the pretreatment of SWRO desalination processes, achieving a reduction of more than 50% of total organic carbon and 90% of microorganisms with a consumption of 0.5 kWh/m$^3$ [122]. Similarly, Liang et al. [123,124] have reported using metallic glasses (MG) as catalysts in water dissociation, remediation of organic pollutants, oxidation methanol and ethanol hydrogenation, etc. This technology can be used as an effective pretreatment in SWRO desalination processes.

Coagulation after screening with filter material is one of the most-used pretreatment processes. FeCl$_3$ is the leading coagulant thanks to its low seawater solubility. For algae elimination, turbidity reduction and suspended material, dissolved air flotation (DAF) or sedimentation operations are usually carried out. In many cases an ultrafiltration membrane (UF) is used with or without the previous use of granular filter media. The use of UF membranes can eliminate the need to clean the RO membrane with savings of up to 30% in these operations. However, UF membranes are not effective for the removal of dissolved organic substances that can cause biofouling [119], which is why the use of chlorine or sodium hypochlorite is common for eliminating bacterial species. 83% of processed water uses chlorine as a disinfection agent, followed using UV radiation (13.4%), and to a lesser extent chlorine dioxide (3.2%) [14]. It is estimated that the costs associated with the fight against fouling of the membranes reaches 15,000 USD per year [125]. The air flotation dissolved (DAF), microfiltration (MF) and dual media filter (DMF) are the most used pretreatment methods, in individual or combined approaches [14]. Table 4 shows the most

used pretreatment methods in the RO plants installed. Data for Middle East and North Africa plants are not available in the study.

**Table 4.** Typical pretreatment methods in the RO plants installed.

| Pretreatment | Accumulated Capacity (m³/day) |
|---|---|
| Ultrafiltration (UF) | 10,178,509 |
| Dissolved Air Flotation (DAF) | 5,262,871 |
| Dual Middle Filter (DMF) | 5,265,368 |
| Microfiltration (MF) | 2,634,964 |
| Sand Filtration | 1,046,195 |

Taken from [10].

On the other hand, water permeability and salt rejection tend to have inversely proportional relationships in most commercial membranes. Improvements in RO membranes include increased separation performance, increased chemical stability and antifouling properties, cellulose acetate membranes, and more recently aromatic polyamide membranes, which have been very well received in the industry [126]. Research is focused on mixed RO membranes. Doping and functionalization with organic and inorganic nanoparticles will allow to obtain membranes resistant to extreme conditions such as elevated levels of acidity [127], surface coatings [128], in-situ surface modification [129], incorporation of two-dimensional materials [130], and preparation of grafts [128] and others. Design with hydrophobic or highly branched molecules, together with advances in nanomaterials such as carbon and graphene nanotubes [130], silver nanoparticles, nano clays, and molecular sieves promise a wide range of possibilities for membrane use in RO applications [16].

Manufacture of polyamide membranes with solid nanofillers allows to increase rigidity and density [131]. Incorporation of negatively charged silver nanoparticles improved rejection of salts, and since interfacial distances are greater than the sizes of water molecules, it was also possible to triple water permeability [132]. Both hydrophilic and hydrophobic silica nano-fillers can be used to modify permeability and selectivity of the membranes. The use of polymeric nanospheres of aminophend/formaldehy resins managed to double water flow to stop membranes without nanospheres reaching values of up to 71.3 L/m², maintaining rejection over 96% and improving antifouling characteristics and resistance to chlorine [133]. The use of the nanocomposite $MX_{ene}$ $Ti_3C_2T_x$ achieved increases of 45% in the water permeabilities, with rejections to NaCl of 98.5%. The inclusion of negative charges allowed the rejection of incrustations [134]. The membranes to which hollow mesoporous silica nanospheres were incorporated and modified with amino groups, managed to improve the permeate flux by 18.9%, achieving fluxes of 63.4 Lm²/h, compared to 44.8–63.4 Lm²/h without the incorporation of the nanospheres, in brackish water [135].

Studies carried out using computational fluid dynamics coupled to experimental developments in RO membranes for brackish water, allowed evaluation of the effect of different geometries of the feed spacer, finding that it causes variations of up to 6.83% in specific energy consumption and 10.42% in permeate concentration [136]. Inclusion of this parameter in manufacturers' software may allow a greater diversity of membranes optimized for specific applications.

Ruiz & Nuez [137] studied the impact of different Feed Spacer Geometry (FSG) on membrane performance of SWRO systems with different water permeability coefficients (A). They observed that the longer the pressure vessels, the greater the influence of FSG on SEC. In terms of SEC, membranes with a lower A-coefficient suffered a more profound FSG impact. On the other hand, spacers play a significant role in improving water permeability in membrane desalination processes, but also increase pressure drop and specific energy consumption for fresh water production. The complex coupling of the non-linear



channel flow along with the mass transfer of water and salts through the membrane, makes it difficult to determine an optimal spacer design [138]. The geometric design effect of the spacers of the spiral wound membrane modules (SWM) of RO systems is crucial to improving these systems' performance. The use of current feed spacers in SWM modules is beneficial for improving mass transfer and preventing polarization and concentration scaling but causes faster fouling and biofouling. Therefore, use of just one type of spacer in RO SWM modules for different applications is not beneficial and instead it would be more advantageous to use a specific feed spacer for a particular type of feed water [139].

Although publications on the synthesis of new membranes have been increasing, their performance is rarely reported in plants in operation. Further investigation of the industrial performance of membranes in the long term and with real waters is needed. The use of nanomaterials, although promising, should be taken with caution as the effects they may have on human and environmental health are still unknown. Further understanding is required on the possible synergies of various strategies to increase antifouling properties, increase water permeability and maintain rejections of salt [140]. Multidisciplinary studies will be crucial to move from the ideality of laboratories to the reality of industrial desalination.

RO desalination is a knowledge, technology, and energy intensive process. The increase in desalination capacity has generated concerns about the environmental impacts of these technologies, related to energy consumption [17], disposal of brine, noise pollution and impacts on the marine environment caused during the construction and operation of plants [141].

Brines are usually dispensed as discharges into surface and marine waters or in the sewage system [17], generating important environmental problems in the marine environment due to change in salinity and pH [9], as well as presence of heavy metals and other residual chemicals used in water pretreatment and brine post-treatment stages, such as antiscalant, coagulants, flocculants, and halogenated disinfection by-products [142], which have known toxic effects on aquatic life [143]. Kress [144] presents a compendium of observations and research reported by various sources regarding the effects of desalination processes in the marine environment.

It is estimated that dumping of brines totals 141.5 million $m^3$/day, concentrated in the Middle East and North Africa regions that generate 55% of the brine derived from desalination operations in the world [18,145]. One of the approaches aimed at reducing this problem is Zero Liquid Discharge (ZLD) [9], and recently the Minimum Liquid Discharge (MLD) with which the fresh water recovery can reach values of 2.21 US $/m^3$ [146]. Evaporation tanks are also used with the subsequent disposal of saline solids in sanitary landfills, causing impacts related to changes in land use, energy consumption, and its possible leaching into groundwater [147].

Brine dilution operations have significant energy and economic costs; the Alicante plant in Spain reported energy consumption of one million kWh during six operation months with an associated cost of €88. Therefore, an increase in the minimum dilution threshold of 2 to 1 imposed by regulatory entities represented the consumption of more than 600,000 kWh between January and June 2020, equivalent to 126 tons of $CO_2$ emissions. This case study also offers an example of how the desalination operation performance can be affected due to regulatory and environmental factors since, when monitoring, stations report values higher than 38.3 in the Practical Salinity Units (PSU) in more than 25% of the observations, or 39.5 PSU in more than 5%, as progressive stop protocols must be implemented which reduces plant production [148].

A novel approach to the brine treatment from desalination processes seeks to see them as a resource in brine mining schemes meeting the requirements of the circular economy [149]. Some RO brines may contain lithium ions, widely used for electric batteries. The Doha and Shuwaikh plants in Kuwait contain 1.7 mg/L of lithium, while the geothermal brines of the Imperial Valley in the United States are between 150–400 mg/L. Brine

mining has the advantage of having low exploration and capital costs [150]. Electrochemical processes have been studied for the recovery of valuable elements from brines. Calcite and brucite compounds can be precipitated with the use of pristine graphite anodes and titanium-coated graphite cathodes [151]. The main challenges to overcome are in the energy consumption required for the purification of the salts and recovery of the water from them (31.81 kWh/m³), so it is essential to evaluate the use of residual energy and renewable energy for these operations, as well as conducting market and technical and economic feasibility studies of ZLD systems.

On the other hand, Pressure Retarded Osmosis (PRO) processes have an important potential for energy generation using brines from desalination by RO. Two-stage PRO processes can increase energy generation. Some of the advantages of this emerging technology are that it does not generate hazardous waste, produces energy from salinity gradients, and can be coupled to RO desalination systems. Nonetheless, it is still necessary to advance in membranes that meet the specifications of high pressure and low internal polarization to be able to efficiently obtain energy [152].

Life cycle analysis is a tool that allows for establishing impacts during the construction, operation, maintenance, and closure phase of RO plants or their components [117,153]. The right proposal of regulations and laws requires considering all supply chains to increase desalination sustainability. For the Australian case the electricity sector accounted for 69% of GHG emissions from seawater desalination processes [154]. On the other hand, studies from life cycle analysis for wastewater reuse, rainwater harvesting and SWRO in the context of Florianopolis in Brazil, found that the SWRO had the greatest environmental impacts. In the case of South Africa, the environmental impacts increased since this country has a greater dependence on thermoelectric energies compared to Brazil, which means that life cycle analyzes have a high degree of correlation with the geographical area of study [155]. RO desalination projects require studies to propose strategies to minimize the environmental impacts of the plant, especially those associated with brine discharge. Constant supply of scientific information will allow for removal of false assumptions and making informed decisions to guarantee fresh water supply and protect the environment [156].

## 3. Hybrid Systems in Membrane Desalination Technologies

Hybridization of RO technology has two conceptual fronts. On the one hand, there are options in which the configurations of the RO of batch and semi-batch operations are hybridized to incorporate aspects of each operational approach aiming to improve efficiency, flexibility, reduce size, and increase recovery [157]. Secondly, it is the coupling with different desalination technologies that seeks to reduce weaknesses and increase strengths by generating emerging synergies [111]. Since each of the membrane technologies employs different driving forces and have their own advantages and disadvantages, their hybridization can overcome limitations associated with the physical or economic nature of the separate processes [101], other authors specifically address hybridization from its coupling with different energy sources, especially renewable energies [158–160]. The adequate selection of the hybridization of different technologies leads to the reduction of energy consumption and environmental impacts, increasing the efficiency of desalination processes [161].

On the other hand, cogeneration and multigeneration plants can reduce GHG from 2 to 4 times. The payback time of the investment can be 2.8 times less compared to a power generation plant. Typically, multi-generation plants that provide more than three useful outlets have gained strength in recent years [162]. The integration of processes through cogeneration and multigeneration strategies is one of the alternatives with the best results for optimizing existing systems. Innovations in designs aimed at using waste heat and coupling diverse sources of renewable energy are the methods most used to solve energy and environmental problems of RO [26].

The results of a number of documents published in the 2010–2020 period are presented in Figure 5 using the Scopus database. A growing tenure can be observed with all search factors; reverse osmosis desalination (ROD), its hybridization, the use of renewable energies (RE), and finally, the coupling between the three, (ROD + Hybrid + RE). All search factors have shown increasing trends.

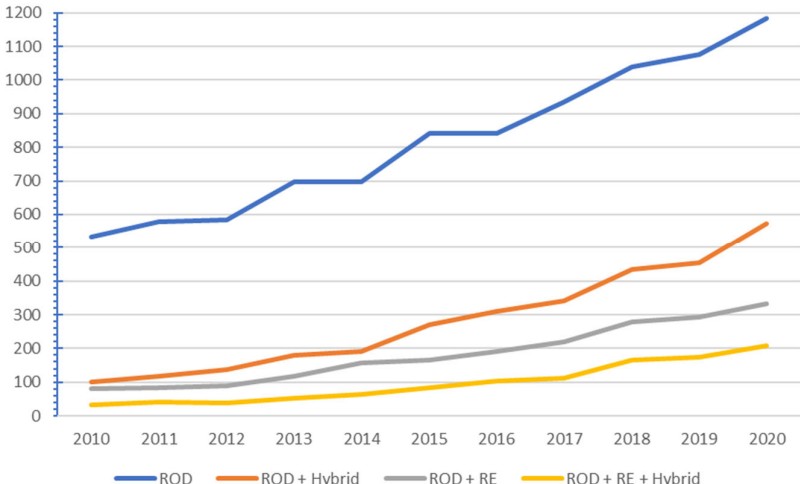

**Figure 5.** Number of publications in the Scopus database per year in the last 10 years for different searches, in which ROD = Reverse Osmosis Desalination and RE = Renewable Energy.

The first RO hybridization studies were presented at the end of the 20th century and consisted of couplings between thermal systems with RO systems that were rapidly gaining popularity and efficiency, combining the benefits of high separations of multistage flash systems, installed around the world, with the increasingly low consumption of RO [161]. MSF-RO hybridization is widely used in plants in the Middle East where low fuel prices have driven the establishment of thermoelectric plants, in which waste thermal energy and low-cost electricity are available. The Al-Fujairah-2 plant in the United Arab Emirates produces 2000 MW of electrical energy, and has used the simple hybridization MED-TVC-RO. It has a combined production of 591,000 $m^3$/day [163]. The MED system (450,000 $m^3$/day) is composed of 5 units, the RO system produces 136,000 $m^3$/day, with a SEC of 3.8 kWh/$m^3$, while the complete system has a SEC of 4.5 kWh/$m^3$, at a cost that varies between 0.6 and 1.6 US \$/$m^3$. At the time it was the largest SWRO plant in the world [14,76].

The Ras Al-Khair plant in Saudi Arabia is currently one of the largest plants in the world that works with this hybridization and produces 1,036,000 $m^3$/day of fresh water, 309,000 $m^3$/day with RO technology, and 727,000 $m^3$/day with MSF, fed with a thermoelectric plant with a capacity of 2400 MW [164]. This allows the RO system to operate with a single stage since the combination of the produced waters meets the quality requirements, and the costs of cleaning and replacement of membranes are reduced. The process can have improvements since the steam collected in the MSF unit can preheat the feed to the RO system increasing its productivity. Because the plant works coupled to a thermoelectric plant, it can produce more water at times of low electricity demand [163]. In general, thermal desalination methods consume about 40 kWh/$m^3$ of electricity, and auxiliary equipment between 2.5 to 5 kWh/$m^3$ [26]. Studies carried out using thermo-economic optimization have found that hybridization with RO increases the efficiency compared to the MED system, with the increase in recovery rates compared to the RO system [165]. In this sense, one way to optimize hybrid MSF-RO systems is through computational modeling and developing advanced simulation algorithms for desalination [166].

Due to this hybridization potential, the effect of different superstructures on key process variables has been studied. Two simple hybridization configurations are assessed in which seawater is fed directly to the MED-TVC and RO system (in a single block and with reprocessing of permeate) and both permeate, and distillate are brought together in a sole product stream: one configuration in which the MED-TVC system is upstream with respect to the RO system; i.e., the MED rejected brine is fed to the RO system, and finally another in which the RO system is upstream with respect to the MED-TCV system (Figure 6), which presented the best performance metrics with salinities lower than 41 kg/m3, average energy consumption of 14.51 kWh/m3, 36.03% water recovery, and the best productivity (91.80 kg/s of fresh water); a sensitivity analysis with the most relevant quality parameters. The amount of vapor used in the MED system influences the quality and quantity of the product in an important way because in this configuration the MED process contributes 75% of the production of the system. The MED configuration in upstream is the most sensitive to salinity [167]. Simulations recently carried out by Abubak et al. [168], confirmed the high performance of MED-TVC coupled with RO, by generating higher productivity and total water recovery, lower specific energy consumption, and a competitive reduction in brine flow. Figure 6 shows the diagram of a RO-MED-TVC hybrid system.

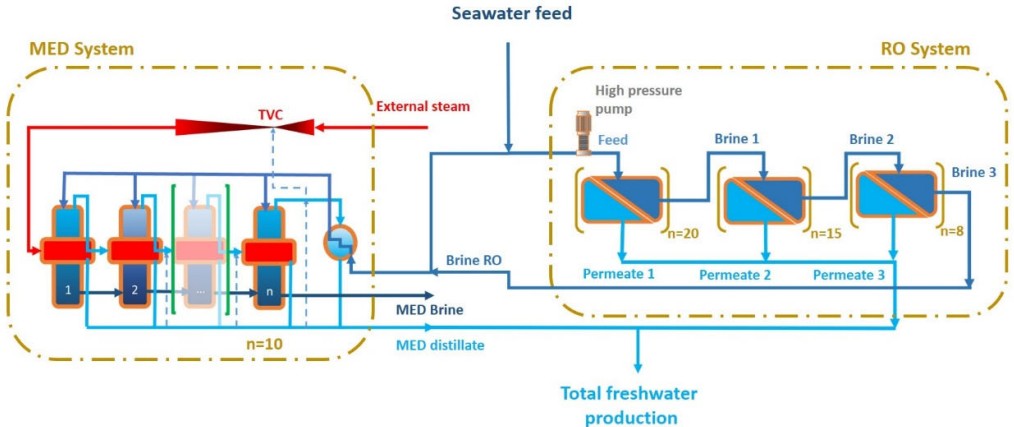

**Figure 6.** Diagram of RO-MED-TVC hybrid system, the RO System is upstream, proposed by Filippini et al. [167].

A cost evaluation with optimization of this proposal was carried out by Al-Obaidi et al. [169]. Minimization of costs allowed obtaining fresh water at a cost of 0.66 US \$/m3. The system exhibits a low sensitivity to salinity and temperature, with a high influence on the electricity cost and therefore on the plant location; a 50% increase in the electricity price produces a 22% increase in fresh water cost, while the same increase in the vapor price produces a 7% variation in the price of fresh water.

In a later work, a photovoltaic (PV) energy source was incorporated into the optimized model. In this case, four separate locations were proposed, the energy of the PV panels is stored in lithium battery packs to guarantee continuous production, seawater goes first to the RO units, and the rejected brine is mixed with more seawater to carry out a MED desalination process (10 stages and operating temperature of 66 °C). Energy production cost was estimated between 0.06–0.15 €/kWh while fresh water costs were 0.55–0.47 €/m3 according to the site and its climatic conditions. For the Isola da Pantelleria location in Italy, a reduction of 34% was achieved compared to the option without renewable energies. The same occurred with the Palmas de Gran Canarias location (22% reduction), but not with the Abu Dhabi location (UAE) where the low price of fuel results in a lower price (0.54 €/m3) compared to the proposed option (0.61 €/m3). Finally, in Perth, Australia, the inclusion of wind turbines achieves more competitive prices (0.47 €/m3) than

the exclusive proposal of PV panels (0.54 €/m³) [170], confirming that location has an important influence on the economic viability of projects powered by renewable energies.

Al-hotmani et al. [81] studied nine designs, including the four designs evaluated by Filippini et al. [167], in which simple and serial hybridizations of MED-TVC and RO technologies were carried out. Some of which incorporate permeate reprocessing in multiple RO units with and without adding an ERD to deliver part of the contained energy in the retention of the first two RO blocks towards the feeding of the third stage. It is highlighted that all configurations that reprocessed the permeate in multiple RO units in series achieved superior indicators of water quality, with reductions in the salinity of fresh water compared to Filippini's proposals. In this case the simple hybridization (Figure 7) proposal incorporating an ERD unit (80% efficiency) manages to reduce energy consumption by 2.2% compared to the same option without a recovery unit, with a significant improvement in the product quality (10,882 ppm vs 141,500 ppm). Equivalent results are observed when comparing this configuration with the best option presented by Filippini et al. [170] with a 3.9% reduction in energy consumption and improvements in fresh water quality (10,882 ppm vs 144,246 ppm). The authors recommend conducting pilot tests with permeate reprocessing designs in the proposed configuration to understand and improve this type of hybridizations and achieve significant improvements in large-scale processes. Figure 7 shows a diagram of MED-TVC-Ro coupled with an ERD unit.

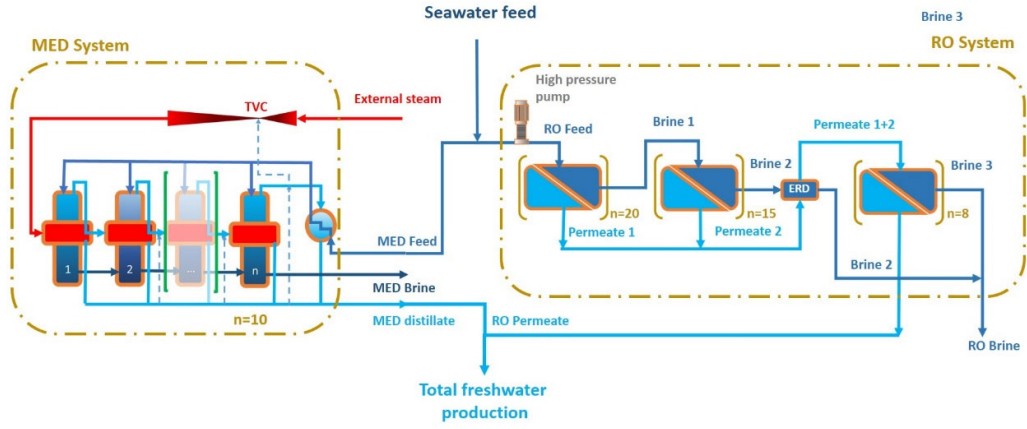

**Figure 7.** Diagram of MED-TVC-RO simple hybrid system with ERD unit proposed by Al-hotmani et al. [81].

Since the obtained configuration had restrictions on the operating variables, the authors subsequently carried out an optimization exercise modifying some operating variables using a non-linear programming (NLP) that sought to minimize the SEC, achieving reductions of 18.26% (14,296 kWh/m³ for the base case to 11,685 kWh/m³ for the optimized case). An increase of 21.8% in the production of fresh water was also reported. The use of a high-pressure membrane allowed operation at 59.9 atm, with increases in flow processed, significant improvements in water quality (87.6%) and increases in recovery (from 30.66% to 34.55%). For future studies, an increase in the level of integration with the coupling to a renewable energy source is proposed [94].

In gas power plants, around 70% of the heat is usually wasted in turbine cycles. Using a Heat Recovery Steam Generator (HRSG) it is possible to take advantage of part of this heat to generate fresh water through MED-TVC units, while the electricity generated drives an RO unit. A SPECO thermo-economic analysis was used for stochastic optimization, seeking to minimize production costs using a genetic algorithm method. The estimation of the parameters was carried out using a heuristic methodology. Six configurations with constant and variable production approaches for the MED-TVC unit were considered. It was found that in the configuration in which the seawater is fed to the RO system after having been used as the cooling water of the MED-TVC system, a reduction between

10.5 and 11.5% of the $CO_2$ produced by the system was also achieved [171]. The proposals for preheating the power supply for the RO system allow achieving higher levels of integration and energy efficiency. Before entering the RO unit, seawater was used to reduce the temperature of the photovoltaic panels, increasing their electricity production, while the productivity of RO units was increased [172].

Based on the need to revamp desalination facilities that consume fossil fuels, Moharram et al. [173] carried out the techno-economic analysis of a proposal for integration with renewable energies at the Ras Gharib plant, Egypt, for cogeneration of energy and water; a solar thermal plant (Parabolic Through Collectors, PTC) coupled to a Rankine vapor cycle for electricity generation, which powers the SWRO desalination plant. A MED subsystem in a simple hybrid configuration condenses the steam from the thermoelectric solar plant to generate more fresh water from seawater. Electricity surpluses can be connected to the grid and the mathematical model (MATLAB/Simulink) was validated with information from the MATS CSP experimental plant in Borg el Arab. The location environmental data were used to calculate the power generated by the system depending on the month. The MED plant achieves average costs of 0.442 US $/m$^3$, with a production of 21,000 m$^3$/day, while the RO plant produces 2000 m$^3$/day at a cost of 0.461 US $/m$^3$, competitive values compared to the prices of plants in the region (1 US $/m$^3$). The plant also produces 15.5 MW at a cost of 0.0452 US $/kWh [173].

In another study by the same authors, a proposal modification is presented, using an MSF subsystem (capacity 14,000 m$^3$/day), instead of a MED (20,000 m$^3$/day), informing that since there are no experimental or theoretical references of the proposed model, a validation of the individual components is carried out. Due to this change, a reduction in the total average capacity of the plant (14,054 m$^3$/day), and a higher cost than the previous proposal of 0.487 US $/m$^3$, were reported, production with slight increases compared to the proposal with MED (0.0458 US $/m$^3$). Power production was also reduced (12.65 MWe vs 15.5 MWe in July corresponding to the higher productivity) [174]. The differences in the results may be associated with the higher capital cost, higher specific consumption of thermal and electrical energy of the MSF system (2000 US $/m$^3$/day, 250–350 mJ/m$^3$, 20 kWh/m$^3$), compared to the MED desalination (1600 US $/m$^3$/day, 150–360 mJ/m$^3$, 17 kWh/m$^3$), and at the lowest recovery rate (10–25% for the MSF system and 23–33% for the MED system) [24,175,176].

A plant operation in Iran that uses hybrid RO systems, together with multi-effect distillation (MED-TVC) to produce energy and water driven by solar and wind energy was evaluated. The heat generated in a cylindrical-parabolic collector produces electricity through an Organic Rankine Cycle (ORC), which together with the energy from the wind turbine supply electricity to a RO unit to produce fresh water. A multi-effect distillation system and thermal vapor compression (MED-TVC) uses the residual heat of the condenser to increase the fresh water production. The system is optimized, using a method of swarm particle optimization, to find the optimal size of each component that maximizes the production of fresh water taking full advantage of the exegetical effectiveness of the Rankine cycle, achieving a production of fresh water at a cost of 3.08 US $/m$^3$. This hybrid system achieves a reduction of 52,164 tons of $CO_2$ emissions per year [177].

On the other hand, and to solve the problems of high specific energy consumption and excess harmful ions in the fresh water production from small RO plants, Yao & Ji [178] proposed to couple to an RO, a membrane capacitive deionization (MCDI) system to produce 2 m$^3$/day of fresh water, with a salt concentration of less than 280 mg/L. The results showed that, when comparing this hybrid system with the original RO system, it is possible to reduce the specific energy consumption for the same effluent salt concentration. Further, the pressure of the RO feed water was lowered, thereby reducing the size of the high-pressure pump for small desalination plants with this technology.

To advance in the achievement of the UN sustainable development goal 6 from related to access to drinking water, Egypt has adopted the strategy "Vision Egypt 2030" that seeks to guarantee access to water and energy using hybrid renewable energy generation

systems (HRPGS) coupled to RO desalination plants. It has been reported that achieving an energy system fully powered by renewable energy for the Middle East and North Africa (MENA) region is technically feasible and economically profitable by 2030, especially using wind and solar energy, thus achieving cost reductions between 55 and 69% compared to traditional options [179,180]. The SWRO is the most attractive alternative to achieve water supply drinking water for the MENA region at a level total cost for 2030 of 1.4 €/m³. The region produces 275 million m³/day with SWRO technology and 5 million m³/day with autonomous MED systems. The region high aridity makes technologies based on biomass or hydroelectric energy unfeasible [180], the vertiginous advances in solar photovoltaic and solar thermal energy have increased the interest in researching the integration with RO, especially in the areas with high solar radiation [19].

## 4. Use of Renewable Energies with Membrane Distillation Technologies

By 2016, world electricity consumption reached 20,863 TWh. Most of this was generated from non-renewable sources such as coal (38.3%), natural gas (23.1%), and oil (3.7%), being the largest contributors to GHG. Only 34.9% of the energy comes from other sources, such as hydroelectric (16.6%), nuclear energy (10.4%), waste energy (2.3%) and solar, wind, geothermal, and tidal energy, which combined contributed only 5.6% of the world production [57]. Moreover, a 37% increase in energy demand is expected for the next 30 years, which would imply a competition between energy consumption and water production with desalination [181]. RO has been classified as the most appropriate technology for desalination integrated into renewable energies. Solar and wind farms seem to be the most suitable alternatives for integration with RO, due to its high development state [182]. Nonetheless, the solar, wind, and tidal profiles, as well as the availability of hydroelectric, geothermal or residual thermal energy of each one of the study regions, are fundamental to establish the RO economy driven by renewable energies [158], and to determine which type of desalination technology is the most appropriate for each area [57].

Transition towards sustainable energy is one of the greatest challenges facing society. The use of renewable energy resources introduces a series of challenges mainly to the inherent variability of wind and solar energy systems, the most widely used technologies [183]. The World Bank considers that 99% of the $CO_2$ generated by desalination processes could be avoided if renewable energy sources were used [184], showing the potential of the use of these energies applied to desalination. Improvement in exergetic efficiency in systems that use solar and wind energy is fundamental since, although the sources are free, the equipment used to transform, and store energy has costs associated with their manufacture, operation, maintenance, and final disposal [185]. Thermodynamic studies are a fundamental tool for understanding, improving and optimizing desalination systems. An approach by operating units will allow identifying the most inefficient elements, while global analyzes will serve to identify synergistic elements to be potentiated, especially in systems that employ different desalination technologies, various energy sources [55], as well as multiple products or services integrated in cogeneration, multigeneration and polygeneration plants.

Khan et al. [158] presented an extensive review of the possibilities of integrating renewable energies with RO in small and large-scale plants, with emphasis on conditions in Saudi Arabia, including a discussion of the use of some software tools for the design and optimization of Hybrid Renewable Energy System (HRES) plants, coupled to RO. Development of renewable energy projects in the Middle East with an installed capacity of 18.9 GW stands out, in which Saudi Arabia contributes 92 MW, a value that can be increased to 9.5 GW by 2030. It is reported that photovoltaic energy contributes 43% of the desalination capacity driven by renewable energies, while solar thermal energy follows with 27%, and wind with 20%. Photovoltaic systems in operation have sizes from 0.8 m³/day to 60,000 m³/day, with costs between 34.21 and 0.825 US $/m³, while wind power plants have capacities between 1 and 250,000 m³/day, with costs that vary between 15.75 and 0.66 US

$/m^3$. Finally, the plants that work with a hybrid supply of solar and wind energy have sizes from 3 to 83,000 $m^3$/day, with costs between 6.12 and 1.4 US $/m^3$ [158].

Small-scale desalination systems powered by renewable energies have seen a significant boost in recent years. More than 130 desalination plants have been installed in the world using energy sources such as solar, wind, and geothermal [24]. Bundschun et al. [186], presented a state of the art of the use of renewable energy sources used in different desalination plants around the world. They found in their study that renewable energies can be successfully integrated with various desalination technologies. Photovoltaic panels are the most used technology in RO desalination. Variability of climatic conditions, fouling of surfaces, and heating of panels were identified as the most influential factors in reducing the capacity of this technology. The optimization of the solar systems requires progress in materials with greater capture capacity and self-cleaning capabilities. It is expected that soon the developments and optimization of the use of oceanic thermal energy will achieve industrial capabilities that allow the use of temperature gradients. Geothermal energy is inexpensive and easily integrable with well-developed thermal technologies. Both solar energy and wind energy require greater advances in energy storage capacity.

Solar energy can be used for electricity production with photovoltaic panels, in solar thermal plants or solar chimneys, as well as to produce heat. With the use of thermal collectors, intermittency problems have been addressed with the use of energy storage systems and hybrid systems for the supply of energy to plants [26]. The integration of concentrated solar power plants with desalination technologies is a promising alternative, especially since some places with a high solar incidence tend to have an inverse correlation with the fresh water availability, leading to a natural symbiosis between the two technologies. Prices of electricity generated with this technology have reached record figures of 6.1 and 7.3 USc/MWh for the Dubai plants (700 MW) [187] and from South Australia (150 MW) [188]. There is the possibility of using the residual heat from the Rankine cycle for the desalination processes. This type of operation would achieve a reduction in $CO_2$ emissions of more than 300,000 Tons per year, compared to the consumption of fossil fuels [189]. Figure 8 presents the condensed information of some of the desalination plants powered by photovoltaic energy in different regions of the world.

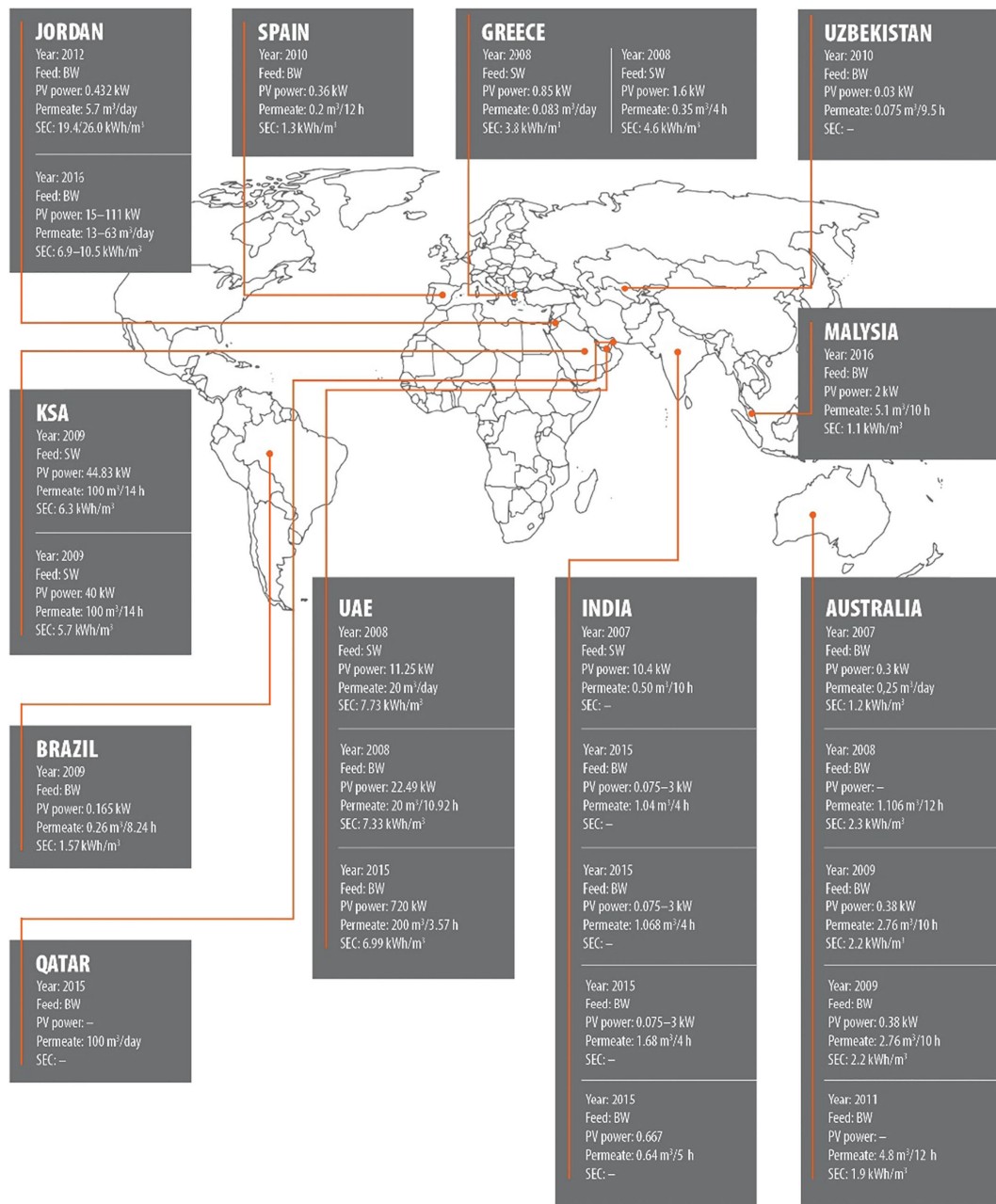

**Figure 8.** Some of the most important desalination plants powered by photovoltaic energy by region. Taken from [186].

Oceanic energy has immense potential due to its abundance, low relative environmental impact [190], high energy density, lower intermittency and predictability compared to other renewable energy sources [191]. It is estimated that it is possible to produce up to 885 TWh of electricity per year in the world [192]. Nonetheless, this is a technology that is in the research and development stages [193], and its use has been limited due to low technological development and the costs associated with collection and energy transport, as well as the maintenance of underwater equipment [194]. Wave and tidal energy can drive RO desalination processes directly by increasing water pressure through mechanical devices or by indirectly generating electricity, subsequently used in an RO unit. It is estimated that wave energy has a potential of up to 80,000 TWh/year [7]. Due to the low development of this technology, questions remain regarding water production costs, influence of seasonality and intermittency of waves, and environmental impacts.

[7]. Environmental considerations must be carefully evaluated since there are still gaps regarding the impact that this technology could have on the marine environment [195]. The study of wave variability on the world's coasts is an input of significant importance for the establishment of large-scale projects and, together with considerations of maintenance, water depth, and distance to users, generates multi-objective problems at the time of determining the suitable site for its installation [7].

The first system that used wave energy (driven by waves) for desalination processes was called Delbupy, it was installed in Puerto Rico in 1982 and managed to produce 1.1 $m^3$/day. The Vizhinjam plant in India was built in 1990, with a capacity of 10 $m^3$/day, and was closed after 21 years of operation. The first project with commercial capacity 150 $m^3$/day, was carried out in Garden Island in 2014 [196]. The use of tidal energy was evaluated to boost an RO system without conversion to electrical energy, using a hydraulic turbine. The system can achieve energy savings between 31 and 41.7% compared to the RO system without integration. If the site development cost rate exceeds 40% of the total cost, the system would generate water at higher prices than a conventional RO system, which means that the selection of the installation site is a decisive factor. It is expected that with the commercial development of tidal energy this proposal can be implemented in the future [197]. Knowledge of behavior of tides and waves of the oceans and seas is essential to measure the technical and economic viability of these plants. Laboratory and bench-scale developments that use these data to simulate real conditions are usually an obligatory step in studies.

Lijon et al. [198], researched the behavior of a commercial RO desalination plant connected to a DC/AC converter system and a variable voltage DC source to simulate power generation using wave behavior data in Kilifi, Kenya. An output potential for the Wabe energy converters (WEC) of maximum 7 kW was estimated, with productions of 536 L/day, through RO units powered by electricity obtained by the WEC system. The increase in WEC units stabilizes the production of energy and water with increases in both costs and energy overproduction. To level costs, it is proposed to reduce the downtime of the system including energy storage or the implementation of a hybrid system with photovoltaic or wind energy. On the other hand, studies carried out to research the feasibility of coupling wave energy to a large RO plant (15,000 $m^3$/day) located in the north of Gran Canaria, showed that most of the devices studied could meet the plant energy needs (19 GWh/year), the uncertainty regarding the commercial requirements of the technologies led to the inclusion of hybrid systems with solar energy, which improved performance for several of the scenarios proposed [196]. Unfortunately, there is no thermodynamic or economic analysis of these proposals, and the cost of fresh water obtained by RO powered by tidal or wave energy is not known. This technology still has many obstacles to overcome.

Other forms of sustainable energy coupled to desalination systems are geothermal and biomass, and although geothermal use is more common in association with thermal technologies [199,200], it has recently been used in cogeneration [186,201,202], trigeneration and polygeneration [203–205] schemes. Geothermal energy is a sustainable, reliable and profitable source, and it is estimated that 10,715 MW were generated in 24 countries in 2010. The United States, the Philippines and Indonesia recorded the highest rate of power generation with this technology in 2016 [206]. The possibility of producing electrical energy in remote areas with geothermal energy supplies, using part of this for RO desalination of both groundwater and wastewater, and using the residual heat from Rankine cycles for RO brine processing and the increase in the production of fresh water are proposals that have been gaining ground, as environmental concerns and commitments to reduce GHG [205]. The additional hydrogen production, as well as of chemicals such as chlorine and sodium hypochlorite from concentrated brines [207] increase the integration levels and improve the environmental and economic indicators of these decentralized processes. On the other hand, the World Bank estimates that around 3.5 million tons of urban solid waste are generated daily in the cities of the world, contributing 3% to 5% of

anthropogenic emissions of GHG. It is expected that by 2025, 3.5 million tons per day will be reached. Biological processes such as anaerobic digestion, and thermal gasification, pyrolysis and incineration processes allow obtaining energy [208] and biofuels such as synthesis gas, biogas, biomethane, biohydrogen, and biohitane (a mixture of hydrogen and methane). Waste incineration is a method used in countries with land limitations, as it reduces or eliminates the need for landfills by reducing waste volume by more than 90% and the recovered heat can be used for power generation that can drive RO desalination systems [209]. Gasification schemes are less widespread, because the generation of dangerous gases such as dioxins and furans, require additional treatment operations, as heavy metals do [210].

The water-energy nexus turns out to be one of the most promising approaches to face the present and future challenges on the shortage of these two essential resources. RO desalination technologies driven by photovoltaic and wind energy are playing a crucial role in this integration. A cost study with electricity generation capacity factors was carried out with data from 81 projects in the Middle East, finding a 20% reduction in energy production costs and a 4% reduction in water desalination costs [211]. Cogeneration systems are increasingly being evaluated for both remote communities and large towns [57].

A comprehensive optimization study for RO desalination systems powered by renewable energies was recently published [57], highlighting system size optimizations, operational optimization, and thermodynamics. Now, advances in life cycle cost analysis, minimization of which has been imposed, together with minimization of costs with restrictions regarding minimum reliability indexes are the dominant characteristics in size optimizations, while minimization of operational costs, costs of electricity, water and emissions, maximization of use of renewable energies, and maximization of production of fresh water, are the trends for operational optimizations. Finally, thermodynamic optimizations employ energy recovery maximization, energy maximization, and exegetical efficiencies, as well as the reduction of the energy consumption of the system. Improvements in RO membranes are highlighted as a key element to achieve better yields, reduce fouling and increase the permeate quality. The extension and improvement of thermodynamic optimization techniques around the destruction of exergy and specific energy consumptions is recommended [57].

### 4.1. Energy Storage

Energy storage is one of the cornerstones for the large-scale implementation of intermittent renewable energies such as solar, wind, and tidal energy. Currently, use of batteries for this purpose is not recommended due to their high capital cost, low lifetime, and environmental loads associated with their construction and disposal, so that hybrid systems that integrate more reliable renewable energies such as geothermal or use residual energy from other thermal processes, are evaluated as an alternative to improve the reliability of the RO that operates with intermittent renewable energies. It is expected that with the reduction of the costs of solar systems [212] and with the future development of new batteries and storage systems [213], there will be a significant increase in the use of photovoltaic and wind energy in desalination processes. Figure 9 presents a diagram of the different alternatives for storing energy obtained from renewable sources used to promote desalination by RO.

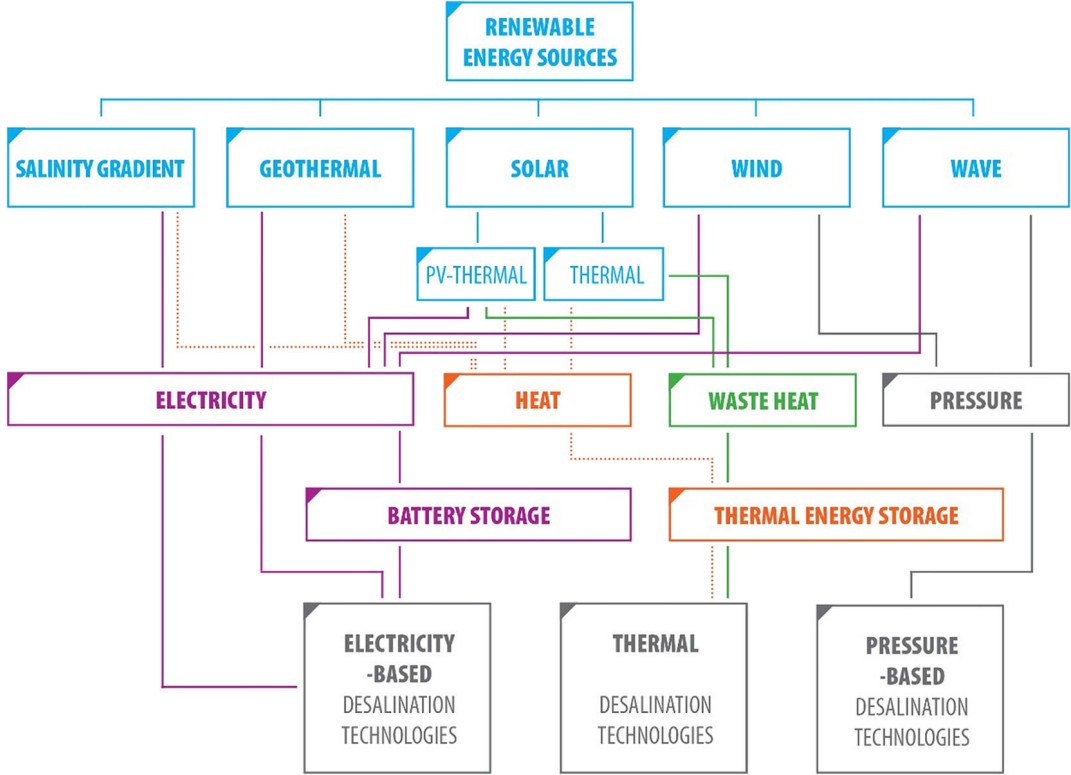

**Figure 9.** Energy storage in processes related to desalination technologies taken from [186].

Multiple energy storage alternatives have been proposed for RO systems that include use of compressed air under water [214] and use of hybrid fuel cells [215]. Water storage tanks have been studied for both reduction of energy consumption through production and distribution management strategies, as well as for storage of hydraulic potential energy [216,217]. The hybrid flow batteries of zinc/ferricyanide have been used for RO desalination processes of hypersaline waters with 85% salt removal and low consumption, 2.11 Wh/L [218]. Seawater batteries (SWB) use sodium ions as a storage system that, along with new developments in desalination with SWB, can drive desalination operations [219]. SWB with electrolytes based on ionic liquids are an emerging technology for storage of electrical energy to large-scale [220]. The use of salinity gradient energy using technologies such as Pressure Retarded Osmosis (PRO), Reverse Capacitive Electro-dialysis (CRED) and Mixing Capacitive (CapMix) is considered an interesting alternative with a high potential, since it uses the salinity difference for the storage and generation of electricity, easily coupled to RO [186].

Although thermal energy is not usually used directly for RO, it can be used for electricity generation or to preheat a system's feed water; some of the alternatives include latent heat storage, with the use of molten salt tanks, water and paraffins [221,222], as well as sensible heat storage using substances such as water, fatty acids, thermal oil, ceramic, concrete, reinforced concrete, refractory bricks, silica, clay, graphite, sodium and iron melts and ionic liquids [186,223]. The integration of different alternatives for energy storage to desalination is a field of study that requires more attention, to overcome the limitations of solar and wind energy, two of the most developed technologies.

### 4.2. Hybridization and Integration of RO for Solar Energy Use

Solar energy has had a growing development in recent decades, but it has been focused mainly on photovoltaic panels, a technology with limitations associated with the fouling and heating of the panels. On the other hand, concentrated solar energy requires

automation systems and specialized monitoring that increase operational and implementation costs [217]. It is estimated that by 2020 there were more than 600 GW of installed capacity in photovoltaic solar panels [19], while by 2050 it is expected that the generation of photovoltaic electricity will achieve a participation between 30% and 50% of worldwide electricity generation [26]. The long useful life, easy maintenance, absence of moving parts and the modular design make photovoltaic energy have several advantages compared to other renewable energy sources [224]. Nonetheless, the intermittent nature of this energy source requires the use of energy storage systems. The first technology used for this purpose corresponded to the use of lead batteries in Saudi Arabia in 1982, with a photovoltaic system that reached an efficiency of 7.5% [225].

Integrated photovoltaic systems with RO desalination have made important advances allowing their commercial implementation. Nonetheless, one of the main challenges in the use of renewable energies is the difficulties derived from their non-continuous nature. It has been shown that the intermittent operation of RO systems increases membrane fouling, and although antiscalant use and membrane rinsing with permeate after each cycle of operation improves the membrane performance, further studies are required to incorporate the variability and intermittence of renewable energies to RO system designs, which consider membrane fouling [114].

Different hybridization configurations for a small-scale MSF-RO system in Tehran, coupled to solar and wind energy was studied by Heidary et al. [226]. Wind speed, meteorological data and solar radiation were input for the determination of energy production. Besides the battery system, the system has storage of fresh water that guarantees the supply when solar and wind power decreases. This system has a greater balance in production capacity than other proposals. However, MSF production is twice the RO (8.3 L/h for the RO system and 16.6 L/h for MSF), the models were optimized minimizing both the cost of energy and fresh water. The configuration that achieved the best results feeds seawater to the MSF system. A part of the produced brine was used as feed to the RO system. Both distillate (from the MSF system) and the permeate (from the RO system) are mixed to form the fresh water flow of the system. The water costs of the optimized system vary between 1.35–1.84 US $/m$^3$, values between 23 and 26% lower than with a single MSF process. MSF performance was improved by reducing RO membrane replacement costs. Increasing RO feed temperature improved its efficiency and the price with wind power at the Al-Karaghouli location was halved compared to the solar system. Furthermore, the reliability of the hybrid system increases since, for the studied location, there is an inverse relationship between wind speed and solar radiation. A sensitivity analysis shows that with changes in the scale the optimal configuration can change, the inclusion of a second MSF unit recovers the heat from the brine from the first unit. Increasing the scale makes the recovery of the MSF brine viable [226].

Cycles of Concentrated Solar Thermal Energy (CST) coupled to electrical generation systems can be useful for the energy supply to hybrid desalination processes. The selection of the configuration of these systems offers a magnificent challenge to designers [227]. The direct solar desalination occurs in devices such as solar stills, in which water is produced directly by solar radiation and despite recent improvements there is still no design that can be used in large-scale installations, but due to its low costs it can be a solution for small rural communities [228].

Indirect solar desalination collects thermal or electrical energy that is later used for the process. Membrane distillation, RO, humidification dehumidification, multi-effect distillation, multi-stage flash, and electro dialysis are some of the main examples [19]. Photovoltaic Thermal Technologies (PVT) are at a high stage of development and research is progressing around the use of the thermal and electrical energy generated [26]. Between 75% and 96% of the solar energy adsorbed by photovoltaic panels is converted into heat and, with increasing temperatures, the efficiency and durability of the panels also decreases. Different proposals have been presented to address this problem, such as the use of hydrogel-based collectors that adsorb water from the atmosphere during the afternoon

and at night and use the residual heat of the panels during the day to evaporate the water by cooling them. This strategy has achieved average cooling capacities of 295 W/m$^2$ on a laboratory scale, reducing temperature of the panels down to 10 °C and increasing their energy generation between 13% and 19% [172].

Since RO increases fresh water generation with increasing temperature, alternatives have been proposed to cool the solar panels while the feedwater is heated to the RO units [229]. Saltwater entry temperature influences the permeate flow and salt rejections, two of the key parameters in the performance of this technology. Temperature increases reduce polarization and reduce the water viscosity, increasing mass transfer phenomena and with them fresh water production. With a pressure reduction required in the operation, it is estimated that for every 1 °C increase in temperature, the pressure required in the system is reduced by 1%. However, temperatures above 45 °C can damage membranes [26]. The design of PVT-RO systems must be careful since reliability must be maintained even in times when there is little solar radiation. Ammous & Chaabene [230] introduce the approaches of dimensioning the area and capacity of the storage tank. Fuzzy logic controllers can offer solutions to the difficulties caused by this type of system, taking care of the integrity of the membranes, maintaining an optimal level of energy production thanks to the cooling of the panels, and guaranteeing fresh water production [231].

The use of RO coupled to Membrane Distillation systems (MD) allows for taking the electrical and thermal energy of the Concentrating Photo Voltaic Thermal (CPVT) collectors. The concentrated brine produced in the RO unit was used as feed to the MD unit, a strategy that increased the production of fresh water and reduced the brine amount, obtaining a recovery rate of 92% and a thermal efficiency of 79.2% [232]. Hybrid systems allow the integrated use of different energy sources, while they can produce fresh water, hot water, and electricity. A small pilot unit, for an isolated house of four people that couples photovoltaic thermal collectors and a vacuum tube collector (ETC) together with a wind microturbine, provides electricity to a RO unit and heat to a desalination unit. The energy storage system used conventional lead acid batteries and a tank for hot water. The experimental values serve to improve and validate a TRNSYS simulation model. It was reported that the plant provides the services reliably. Because all the systems are modular, ease of scaling is estimated [233], the proposed system has an exergetic efficiency of 7.76% (6.68% for electricity, 0.33 for fresh water and 0.75% for hot water) [185]. Desalination through RO driven by photovoltaic thermal collectors reduces the required area by 43.33%, the initial cost by 1.86%, and the required energy consumption by 23.61% [26], being a promising alternative. However, studies are needed around system scaling, maintenance costs and life cycle analysis.

Solar Chimney (SC) technology is one of the oldest technologies used for desalination [234]. It is made up of a collector in charge of heating the space and the ground below it. Warm air rises through a chimney in whose entrance a turbine that generates electricity is installed. The heated floor allows the system to operate even in light absence. Different improvements have been proposed that allow increasing the efficiency of heat up to 17%. Studies agree that the energy produced by this system is related to the height and diameter of the chimney [217,235–237]. Solar chimneys have already been used in desalination systems by means of distillation processes of water [238,239], and its integration with RO is in experimental stages.

Recently Méndez & Bicer [217], proposed a highly integrated hybrid system, composed of a solar chimney, in which the heat stored in the area under the collector is used to drive an MSF desalination system fed with seawater (see Figure 10). The brine produced by the MSF system is fed to an RO plant driven by the electrical energy generated both by the solar chimney turbine and by a 5 × 3.4 MW wind farm, increasing the system reliability. Additionally, the brine from the RO system is used for electricity production with a PRO system, taking advantage of the salinity gradient between it and the seawater. The distilled water from the MSF unit and the permeate (from the RO unit) are mixed and they are pumped into an elevated tank.

The system achieves energy efficiencies of 52.53%, while a solar fireplace dedicated exclusively to power generation only reaches 0.44%. The MSF system fed with 100 kg/s of seawater obtains 8.30 kg/s of fresh water, while the RO system achieves recoveries of 50% at a rate of 45.85 kg/s fresh water, with an energy and exergetic efficiency of 52.7% and 3.13%, respectively. A pressure exchanger manages to reduce the energy consumption required for the pressure increase of the RO system. The PRO system generates 86.11 kW, the combined RO-PRO systems achieve 42.47% energy efficiency and 43.01% exergetic efficiency. The storage systems at height allow increased reliability when storing fresh water and energy potential in times of high productivity, discharge of this water can generate 6.14 kW of electricity [217]. Even though the highly integrated scheme is promising, the authors do not present costs, it is expected that future developments will also allow optimization for a specific location.

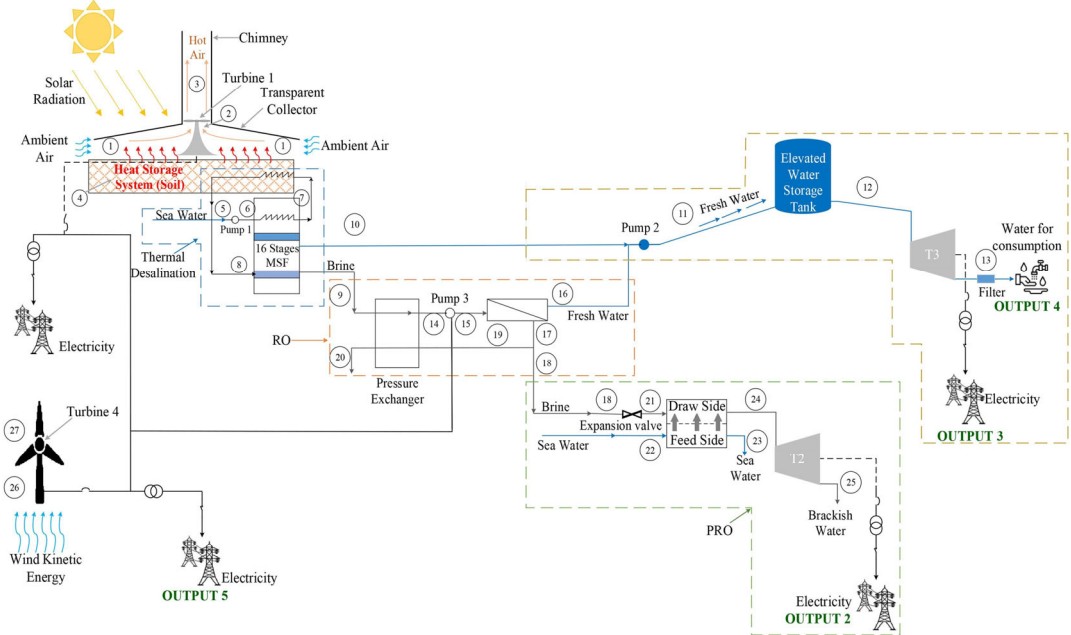

**Figure 10.** Representation of the hybrid MSF-RO system powered by solar energy using a solar chimney and wind energy. Taken from [217].

### 4.3. Use of Artificial Intelligence

Development of Engineering 4.0 will make artificial intelligence an essential tool for the design and efficient management of desalination plants, increasing productivity, reducing errors, optimizing operating costs, and freeing human beings from carrying out repetitive tasks [240]. Energy storage systems and automated production decision-making based on real-time demand analysis will be a fundamental part for the future of the desalination industry [241,242]. Different optimization strategies have been presented for the energy management using fuzzy logic systems [179,243,244], with the incorporation of genetic algorithms [182], Response Surface Methodologies (RSM), Artificial Neural Networks (ANNs), machine learning and expert systems [57]. Modeling of integrated pretreatment processes with RO to maximize water recovery, salt rejection, and reduce energy consumption from different water sources, has been presented with these technologies showing promising results [160].

The solution to the problems derived from the implementation of hybridization proposals highly integrated with renewable energies and with profound complexities will require the increasing participation of non-human intelligences capable of processing substantial amounts of information from diverse sources, a set of technologies that will be more increasingly relevant to water and energy management in the context of renewable

energy-driven desalination. Incorporation of environmental, regulatory, social, and economic indicators in the objective functions will be a requirement for the optimization of desalination systems that seek sustainable solutions to the problem of water supply.

## 5. Future Perspectives

The life cycle analysis of desalination processes is a fundamental tool to establish the environmental impacts of the processes and should be used together with the economic and social assessments to evaluate the impacts associated with energy use and exploration, construction, operation, maintenance, decommissioning, dismantling, and abandonment of desalination plants.

Hybridization is a technique that, through an adequate configuration and operation, allows for reducing weaknesses and increasing strengths of each technology by coupling these hybridizations to renewable energy sources. Research on these aspects has been increasing consistently over the last decade.

RO is considered the technology with the greatest potential for coupling with renewable energies, especially with photovoltaic and wind energy, the two most developed technologies. Profiles of solar radiation, wind speed, tides and waves, as well as other meteorological conditions, are fundamental to establish the feasibility of integrating renewable energies to hybrid desalination systems; the economic evaluations of these alternatives require knowing the location, and demand for water and energy. Several studies have been reported that have proven the thermal and economic viability of these alternatives, for several specific locations, especially the most remote ones.

Several studies have been presented around the use of ocean wave energy (waves) and tidal energy (tides). It has greater reliability due to its lower intermittency, compared to other renewable energies, proving to be a source with immense potential for desalination processes. Nonetheless, this technology is in the process of basic development and there are still no studies to prove its economic feasibility.

Other forms of energy with immense potential are biomass and geothermal energy;most of the proposals are more associated with desalination projects with thermal technologies. However, several researchers have presented hybridization schemes of these with RO involving technologies of cogeneration, trigeneration, and polygeneration, with the simultaneous production of energy, water and products such as hydrogen, biohythane, biogas and biomethane. The use of brine to obtain chlorine and sodium hypochlorite improved the environmental and economic indicators and is an alternative application that integrates the circular economy with the ZLD and MLD proposals.

Because many thermal desalination plants are in places with high solar radiation, the articulation of these facilities with photovoltaic energy sources and solar thermal energy has the merit of allowing a transition to renewable energy using part of the existing facilities. MED, MD, and MSF systems can use thermal energy from thermoelectric or thermosolar plants, while RO uses electrical energy for desalination processes. Coupling thermal desalination systems with RO can also take advantage of waste heat from industrial processes, cogeneration systems, or photovoltaic solar panels.

For the use of thermal energy in RO hybrid systems, the use of MED desalination is more advisable than MSF desalination due to its greater efficiency derived from low operating temperatures, lower energy consumption, low capital cost, and higher distillate recovery. However, due to the fact that the MSF technology is in operation occupying 18% of the installed capacity, several authors have studied a hybridization proposal around this technology. MD's low sensitivity to salinity makes it a perfect candidate for hybridization as it can process RO brines in ZLD or MLD schemes, improving energy efficiency with high recovery yields.

Because the rise in the temperature of the inlet water in the RO units increases the permeate production, preheating systems can be implemented using low-grade waste heat from various processes, and geothermal energy. Cooling the photovoltaic solar panels with feed water raises their temperature while it increases electricity.

Generally, RO hybridization proposals with thermal desalination technologies coupled to renewable energy sources (RES), include thermal and/or electrical energy generation units and desalination systems in simple hybridizations (in parallel) or in series. Incorporation of energy storage systems and fresh water, though can increase costs, usually increase the system reliability by allowing fresh water supply at times when RES reduce their energy production.

Since RO efficiency is associated with the feedwater temperature, hybridizations have been proposed using residual heat from thermal energies and renewable energy sources to preheat the feedwater, increasing the efficiency, and production of the system.

Finally, a case was presented that uses solar chimneys along with wind energy in integration with hybrid desalination systems, and energy recovery systems, with thermal storage, potential energy, and produced water that increase the process reliability. Increases in energy efficiency versus exclusive use of energy show an interesting path for old solar technology.

### 6. Conclusions

Desalination of sea and brackish water has shown a significant increase in recent decades because of the rise in demand for fresh water, driven both by population and economic growth, as well as by deterioration of conventional water sources. RO is the most advanced technology used in the world for desalination due to its relatively low energy consumption (SEC), high efficiency, flexibility, ease of operation, and process economy. Factors that have driven RO include advances in new membrane materials, improvements in high pressure pump efficiency, and introduction and development of energy recovery devices (ERD). On the other hand, polarization increase, low boron retention, membrane permeability loss and their fouling, and the high operating pressures required for desalination from sources of high salinity are some of the technical problems. While brine disposal, noise pollution, impacts on the marine environment, and generation of GHG associated with the energy source are some of the environmental difficulties of this technology.

Research in new membranes development and in the hydrodynamics process will generate solutions to some of the RO technical problems. Coupling with renewable energies and implementation of strategies of zero liquid discharge (ZLD) and minimal liquid discharge (MLD), along with use of brines to obtain valuable products applying circular economy concepts, will be crucial to reduce environmental impacts. A multidisciplinary vision will be crucial both for developing of laboratory-scale research and for the industrial implementation of the proposals derived from these. Research is also required around the development of ERD for small-scale RO.

**Author Contributions:** writing-original draff preparation, J.J.F.-D. and F.C.-M.; writing-review and editing, F.C.-M.; Conceptualization, M.C.L.-M., J.P.R.-M. and J.B.-R. All authors have read and agreed to the published version of the manuscript.

**Funding:** This research received no external funding.

**Institutional Review Board Statement:** Not applicable.

**Informed Consent Statement:** Not applicable.

**Data Availability Statement:** Not applicable.

**Conflicts of Interest:** The authors declare no conflict of interest.

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
