# Peer review of "Recent Desalination Technologies by Hybridization and Integration with Reverse Osmosis: A Review"

_water, doi:10.3390/w13101369_

Round 1

Reviewer 1 Report

The review paper entitled “Recent Desalination Technologies by Hybridization and Integration with Reverse Osmosis: A Review” and written by Jhon Jairo Feria-Díaz et al. is interesting and addresses the hybridization of reverse osmosis with other technologies in terms of desalination and its integration to be powered by renewable energy sources. This review covers two very broad topics on which many articles have been published. Naturally, it is complex to take into account all the published manuscripts related to the subject. Overall the document is well written but the structure could be improved. I recommend a major revision based on the following comments:

  1. Regarding the structure of the manuscript in terms of section and subsections. Why the RO technology (section 3) is out of section 2 (Desalination technologies with membranes). RO is a desalination technology with membranes so this does not make sense to me. Section 3 should be section 2.1 or 2.5, included in section 2 as well as the subsections 3.1 and 3.2. If the authors want to keep the actual structure they should change the title of section 2 by including “alternatives to RO”.
  2. I do not understand why there is a section for membrane based desalination technologies and not one for thermal based desalination processes. One of the possibilities described deal with MSF-RO systems. there should be a section for both type of technologies.
  3. The section 4 is confusing. It is entitled “Technology hybridation of RO” (there is a typo in the manuscript, it is written hibridation) but, the subsections 4.1 and 4.3 are related with desalination powered by renewable energy sources. Please, restructure the sections. If section 4 is about Technology hybridation of RO, join sections 4 and 4.1 and create a new section related with hybrid systems with RO powered by renewable energy sources, this new section should include subsections 4.1 and 4.3.
  4. Please, add the technology in the caption of table 1.
  5. Reverse osmosis should be abbreviated in page 2 line 78 and please, use the abbreviation in the rest of the manuscript.
  6. Page 2, line 59. Authors mentioned that seawater and brackish water desalination are technologies. Actually are processes that can be carried with RO technology for example. I suggest to write processes instead of technologies.
  7. Please, write seawater in the entire document instead of sea water (Page 9, line309, etc…)
  8. Greenhouse gas was abbreviated in page 3 line 111. Please use the abbreviation in the entire document (page 4, line 135, etc…). Please check all the abbreviations.
  9. In table 2, the word membranes should be removed from “RO membranes”. Why Nanofiltration is not included in Table 2? It should be and also described.
  10. Page 8, line 297. There are desalination plants with a larger capacity than 395,000 m^3/d. Please, check the bibliography.
  11. Page 9, lines 312-318. Use units of the international system instead of psi.
  12. Page 10, lines 393-405. I would like to remark a few sentences that should be clarify or modified:
    1. It was mentioned that in configurations with 2 or more stages, “Because the difference in osmotic pressures across the membranes is reduced, different hydraulic pressures can be applied causing a reduction in the SEC

This is not true, increasing the number of stages without inter-stage pump increases the flux recovery of the RO system which decrease the SEC. The difference in terms of net driving pressure or osmotic pressure gradient is not the reason. The feed pressure decreases from one stage to other unless an inter-stage pump is installed and the osmotic pressure of the feed-brine solution increasing as the solute become more concentrated as well as the concentration in the permeate is higher in the second stage than in the first stage. Authors should clarify this point.

  1. Authors mentioned that two-stage configuration became popular at the beginning of the century. The authors should mention if they are writing about seawater or brackish water reverse osmosis. It seems that it is about seawater because two stage configuration for brackish water has been applied from 80’s.
  2. In this section, the authors should mention the relevance of the RO system configuration on the optimization of the process, I recommend the following papers to be commented and cited:
    1. A computational tool for designing BWRO systems with spiral wound modules
    2. Scope and Limitations of Modelling, Simulation, and Optimisation of a Spiral Wound Reverse Osmosis Process-Based Water Desalination
  • A design method of the RO system in reverse osmosis brackish water desalination plants (calculations and simulations)
  1. Optimum design of reverse osmosis system under different feed concentration and product specification
  2. Optimal design and operation of reverse osmosis desalination process with membrane fouling
  3. Simulations of BWRO systems under different feedwater characteristics. Analysis of operation windows and optimal operating points
  • Multi-objective optimization of reverse osmosis networks by lexicographic optimization and augmented epsilon constraint method
  1. The section 3.2 deals with the Problems and limitations of RO technology, of course there are many published manuscript about it and it is hard to consider every single one. The authors should mention that scaling of less soluble salts is one of the main limiting factors in terms of flux recovery (please, check the paper entitled “Estimation of maximum water recovery in RO desalination for different feedwater inorganic compositions”). Other limitation of the RO technology is to reach acceptable boron concentration in the permeate according with the regulation of each country, there are countries where the limit concentration is 1 mg/L and if the permeate is for agricultural irrigation purposes there are crops very sensitive to boron concentrations requiring less than 1 mg/L. I recommend the following papers to be included:
    1. High boron removal polyamide reverse osmosis membranes by swelling induced embedding of a sulfonyl molecular plug
    2. Different boron rejection behavior in two RO membranes installed in the same full-scale SWRO desalination plant
  2. In the same section (3.2) the authors mentioned that fouling is one of the main problems and it decrease the permeability in long term operation, this is correct but the authors should include a published manuscript that show the long term permeability decay due to fouling. I suggest the following:
    1. Impact of membrane ageing on reverse osmosis performance – Implications on validation protocol
    2. Long-term performance decline in a brackish water reverse osmosis desalination plant. Predictive model for the water permeability coefficient
  3. Authors also mention how intermittent operation can affect the performance of RO systems. Regarding this topic there are also missing papers that are relevant in this field:
    1. Impact of intermittent operation on reverse osmosis membrane fouling for brackish groundwater desalination systems
    2. Long-term intermittent operation of a full-scale BWRO desalination plant
    3. Experimental quantification of the effect of intermittent operation on membrane performance of solar powered reverse osmosis desalination systems
  4. Page 14, lines 549-554. Authors mentioned a work that deals with the impact of feed spacer geometries on RO membrane performance. Feed spacer geometries is an important factor that should be considered in the optimization of the RO system It is a success that the authors have mentioned it. Authors also should mention other works that deals with this topic that are relevant such as:
    1. Performance Assessment of SWRO Spiral-Wound Membrane Modules with Different Feed Spacer Dimensions
    2. Optimal design of spacers in reverse osmosis
    3. Sensitivity analysis and gradient-based optimisation of feed spacer shape in reverse osmosis membrane processes using discrete adjoint approach
  5. Page 14 line 581. I think it should be written 2.21 US$/m^3 instead of US$2.21 m^3. Same in line 587, it should be 88,000 €. Please, check the entire document. Page 17, line 730 MW instead of mW right?
  6. Definitely, I need to revise this manuscript with the new structure. Although the number of citations seems appropriate for this type of manuscript, the review is about such broad topics and on which so much work has been done that it requires that they be extended. In addition to the suggestions made above, I recommend that the authors review the following works and consider including them in the revised manuscript:
    1. Modelling and Optimisation of Multi-Stage Flash Distillation and Reverse Osmosis for Desalination of Saline Process Wastewater Sources
    2. A Small RO and MCDI Coupled Seawater Desalination Plant and Its Performance Simulation Analysis and Optimization
    3. An Innovative Design of an Integrated MED-TVC and Reverse Osmosis System for Seawater Desalination: Process Explanation and Performance Evaluation

Author Response

Dear Reviewer.

Thank you very much for the observations and contributions made to our work. In the attached file you will find each and every one of the responses to the observations made.

Reviewer 2 Report

Feria-Díaz et al. present the review particularly focusing on desalination by reverse osmosis, with related hybridization techniques. The topic is attractive, and I have read the text with great interest. The article has shown many relevant data for convincing readers, especially the most concerned issues i.e. energy consumption. It is in general clear and well written. I have some suggestions that can be considered to be clarified in this report, please see below:

  1. If possible, the manufacturing process of the general RO membranes can be included in the review.

  1. Regarding the fouling of RO membrane in section 3.2 Limitations and Problems, maybe a further detailed discussion would be attractive to readers to solve the current problems and consider the further developments.

  1. When considering the pretreatment of SWRO desalination processes, advanced oxidation processes (AOPs) would be one of promising techniques for reducing the total organic carbon. The following papers would be suggested to discuss [Advanced Materials 30 (2018) 1802764; Applied Materials Today 19 (2020) 100543].

  1. What is the perspective and future challenges of RO technique?

Author Response

(The authors gave the same response as above.)

Reviewer 3 Report

See attached comments

Author Response

Dear Reviewer.

We appreciate the time and inconvenience caused during the review of our work, but we do not agree with many of the comments and suggestions made. Therefore, in a respectful way, the authors inform you that we will not take them into account in the new version of the paper.

Sincerely,

The authors.

Round 2

Reviewer 1 Report

The paper has been considerably improved and the authors have addressed all my comments